# The forecasted prevalence of comorbidities and multimorbidity in people with HIV in the United States through the year 2030: A modeling study

Keri N. Althoff[1]*, Cameron Stewart[1], Elizabeth Humes[1], Lucas Gerace[1], Cynthia Boyd[1,2,3], Kelly Gebo[4], Amy C. Justice[5,6], Emily P. Hyle[7,8], Sally B. Coburn[1], Raynell Lang[9], Michael J. Silverberg[10,11], Michael A. Horberg[12], Viviane D. Lima[13], M. John Gill[9], Maile Karris[14], Peter F. Rebeiro[15], Jennifer Thorne[16], Ashleigh J. Rich[17], Heidi Crane[18], Mari Kitahata[18], Anna Rubtsova[19], Cherise Wong[20], Sean Leng[2], Vincent C. Marconi[21,22], Gypsyamber D'Souza[1], Hyang Nina Kim[18], Sonia Napravnik[23], Kathleen McGinnis[6], Gregory D. Kirk[1,4], Timothy R. Sterling[24,25], Richard D. Moore[26], Parastu Kasaie[1]

1 Department of Epidemiology, Johns Hopkins Bloomberg School of Public Health, Baltimore, Maryland, United States of America, 2 Division of Geriatric Medicine and Gerontology, Department of Medicine, Johns Hopkins School of Medicine, Baltimore, Maryland, United States of America, 3 Department of Health Policy and Management, Johns Hopkins Bloomberg School of Public Health, Baltimore, Maryland, United States of America, 4 Division of Infectious Diseases, Department of Medicine, Johns Hopkins School of Medicine, Baltimore, Maryland, United States of America, 5 Yale Schools of Medicine and Public Health, New Haven, Connecticut, United States of America, 6 Veterans Affairs Connecticut Healthcare System, West Haven, Connecticut, United States of America, 7 Harvard Medical School and the Division of Infectious Diseases, Massachusetts General Hospital, Boston, Massachusetts, United States of America, 8 Harvard University Center for AIDS Research, Boston, Massachusetts, United States of America, 9 Department of Medicine, University of Calgary, Calgary, Canada, 10 Division of Research, Kaiser Permanente Northern California, Oakland, California, USA and Department of Health Systems Science, Kaiser Permanente Bernard J. Tyson School of Medicine, Pasadena, California, United States of America, 11 Department of Epidemiology and Biostatistics, University of California San Francisco, San Francisco, California, United States of America, 12 Mid-Atlantic Permanente Research Institute, Kaiser Permanente Mid-Atlantic Permanente Medical Group, Rockville, Maryland, United States of America, 13 Epidemiology and Population Health Program, British Columbia Centre for Excellence in HIV/AIDS, Vancouver, Canada, 14 Department of Medicine, University of California San Diego, San Diego, California, United States of America, 15 Departments of Medicine and Biostatistics, Vanderbilt University School of Medicine, Nashville, Tennessee, United States of America, 16 Department of Ophthalmology, Wilmer Eye Institute, Johns Hopkins University School of Medicine, Baltimore, Maryland, United States of America, 17 Department of Social Medicine, University of North Carolina, Chapel Hill, North Carolina, United States of America, 18 Division of Allergy and Infectious Diseases, Departments of Medicine and Epidemiology, University of Washington, Seattle, Washington, United States of America, 19 Department of Behavioral, Social, and Health Education Sciences, Emory University Rollins School of Public Health, Atlanta, Georgia, United States of America, 20 Division of Worldwide Research and Development, Pfizer Inc., New York City, New York, United States of America, 21 Division of Infectious Disease, Emory School of Medicine, Atlanta, Georgia, United States of America, 22 Atlanta Veterans Affairs Health Care System, Decatur, Georgia, United States of America, 23 Department of Medicine, University of North Carolina at Chapel Hill, Chapel Hill, North Carolina, United States of America, 24 Vanderbilt Tuberculosis Center, Vanderbilt University School of Medicine, Nashville, Tennessee, United States of America, 25 Division of Infectious Diseases, Department of Medicine, Vanderbilt University School of Medicine, Nashville, Tennessee, United States of America, 26 Division of General Internal Medicine, Department of Medicine, Johns Hopkins School of Medicine, Baltimore, Maryland, United States of America

* kalthoff@jhu.edu



**Data Availability Statement:** HIV Surveillance data was sourced from the US Centers for Disease Control and Prevention's HIV Surveillance Reports, available at: https://www.cdc.gov/hiv/library/reports/hiv-surveillance.html NA-ACCORD data is available following approval from the collaboration,

available at: https://naaccord.org/collaborate-with-us Methodological details regarding the model structure, parameterizations, standards for collapsing subgroups to ensure adequate sample size, and estimated functions are available at: https://pearlhivmodel.org/method_details.html The code for the PEARL model results presented here can be found at: https://github.com/PearlHivModelingTeam/comorbidityPaper.

**Funding:** The presented research was supported by the National Institutes of Health (R01 AG053100 to KNA; U01AI069918 to KNA and RDM; K01AI138853 to PK; R01AG069575 to EPH) and the Jerome and Celia Reich Endowed Scholar Award (to EPH). The funders had no role in study design, data collection and analysis, decision to publish, or preparation of the manuscript.

**Competing interests:** KNA serves on the scientific review board for TrioHealth Inc and as a consultant to the All of Us Research Program. MJG has been an Hoc member on national HIV Advisory Boards of Merck, Gilead and ViiV. CW is currently employed by Regeneron Pharmaceuticals Inc and contributed to this article as a prior trainee of Johns Hopkins University. KG declares that his institution receives funding from U.S. Department of Defense's (DOD) Joint Program Executive Office for Chemical, Biological, Radiological and Nuclear Defense (JPEO-CBRND), in collaboration with the Defense Health Agency (DHA) (contract number: W911QY2090012), Bloomberg Philanthropies, State of Maryland, NIH National Center for Advancing Translational Sciences (NCATS) U24TR001609, Division of Intramural Research NIAID NIH, Mental Wellness Foundation, Moriah Fund, Octapharma, HealthNetwork Foundation, and the Shear Family Foundation for her work. KG received royalties from UptoDate and served as a paid consultant to Aspen Institute, and Teach for America. KG declares that none of these funding sources are related to this manuscript. PFR declares consultation with Gilead & Janssen pharmaceuticals (money paid to individual); research grants from NIH/NIAID (money paid to institution). JT declares to be consultant for AbbVie, Canfield, Gilead, Roche and Tarsier and being Equity owner for Tarsier. VFM has received support from the Emory CFAR (P30 AI050409) and received investigator-initiated research grants (to the institution) and consultation fees (both unrelated to the current work) from Eli Lilly, Bayer, Gilead Sciences, and ViiV. HNK declares that Gilead Sciences program funding paid to the author's institution. The following authors have declared that no competing interests exist: CS, EH, LG, CB,

# Abstract

## Background

Estimating the medical complexity of people aging with HIV can inform clinical programs and policy to meet future healthcare needs. The objective of our study was to forecast the prevalence of comorbidities and multimorbidity among people with HIV (PWH) using antiretroviral therapy (ART) in the United States (US) through 2030.

## Methods and findings

Using the PEARL model—an agent-based simulation of PWH who have initiated ART in the US—the prevalence of anxiety, depression, stage ≥3 chronic kidney disease (CKD), dyslipidemia, diabetes, hypertension, cancer, end-stage liver disease (ESLD), myocardial infarction (MI), and multimorbidity (≥2 mental or physical comorbidities, other than HIV) were forecasted through 2030. Simulations were informed by the US CDC HIV surveillance data of new HIV diagnosis and the longitudinal North American AIDS Cohort Collaboration on Research and Design (NA-ACCORD) data on risk of comorbidities from 2009 to 2017. The simulated population represented 15 subgroups of PWH including Hispanic, non-Hispanic White (White), and non-Hispanic Black/African American (Black/AA) men who have sex with men (MSM), men and women with history of injection drug use and heterosexual men and women. Simulations were replicated for 200 runs and forecasted outcomes are presented as median values (95% uncertainty ranges are presented in the Supporting information).

In 2020, PEARL forecasted a median population of 670,000 individuals receiving ART in the US, of whom 9% men and 4% women with history of injection drug use, 60% MSM, 8% heterosexual men, and 19% heterosexual women. Additionally, 44% were Black/AA, 32% White, and 23% Hispanic. Along with a gradual rise in population size of PWH receiving ART—reaching 908,000 individuals by 2030—PEARL forecasted a surge in prevalence of most comorbidities to 2030. Depression and/or anxiety was high and increased from 60% in 2020 to 64% in 2030. Hypertension decreased while dyslipidemia, diabetes, CKD, and MI increased. There was little change in prevalence of cancer and ESLD. The forecasted multimorbidity among PWH receiving ART increased from 63% in 2020 to 70% in 2030. There was heterogeneity in trends across subgroups. Among Black women with history of injection drug use in 2030 (oldest demographic subgroup with median age of 66 year), dyslipidemia, CKD, hypertension, diabetes, anxiety, and depression were most prevalent, with 92% experiencing multimorbidity. Among Black MSM in 2030 (youngest demographic subgroup with median age of 42 year), depression and CKD were highly prevalent, with 57% experiencing multimorbidity. These results are limited by the assumption that trends in new HIV diagnoses, mortality, and comorbidity risk observed in 2009 to 2017 will persist through 2030; influences occurring outside this period are not accounted for in the forecasts.

## Conclusions

The PEARL forecasts suggest a continued rise in comorbidity and multimorbidity prevalence to 2030, marked by heterogeneities across race/ethnicity, gender, and HIV acquisition risk subgroups. HIV clinicians must stay current on the ever-changing comorbidities-specific

ACJ, EH, SC, RL, MJS, MH, VL, MK, AJR, HC, MK, AR, SL, GDS, SN, KMG, GDK, TRS, RDM, PK.

**Abbreviations:** ART, antiretroviral therapy; Black/AA, Black/African American; BMI, body mass index; CKD, chronic kidney disease; CVD, cardiovascular disease; ESLD, end-stage liver disease; HCV, hepatitis C virus; MI, myocardial infarction; MSM, men who have sex with men; MWID, men who injected drugs; NA-ACCORD, North American AIDS Cohort Collaboration on Research and Design; ppc, percentage point change; PWH, people with HIV; RAS, renin-angiotensin system; ROS, reactive oxygen species; SDoH, social determinants of health; US, United States; WWID, women who injected drugs.

guidelines to provide guideline-recommended care. HIV clinical directors should ensure linkages to subspecialty care within the clinic or by referral. HIV policy decision-makers must allocate resources and support extended clinical capacity to meet the healthcare needs of people aging with HIV.

## Author summary

### Why was this study done?

- Individuals with HIV are aging and are experiencing an increased risk of comorbidities and multimorbidity.

- Anticipating the healthcare needs of an aging population with HIV is crucial for healthcare providers, policymakers, and public health officials to plan for medical and support services tailored to the unique needs of aging people with HIV.

- In response to the increasing medical complexity among aging adults with HIV, we employed an agent-based simulation model to forecast the potential burden of comorbidity and multimorbidity in individuals who have initiated antiretroviral therapy (ART) in the US.

### What did the researchers do and find?

- PEARL is an agent-based simulation of persons with HIV who have initiated ART in the US, utilizing data from the CDC HIV surveillance system and comorbidity risk functions derived from the North American AIDS Cohort Collaboration on Research and Design (NA-ACCORD), the largest longitudinal cohort of people with HIV who have linked into care in the US.

- Following a gradual increase in number of people with HIV receiving ART from 2020 to 2030, PEARL forecasted a population of 908,504 in the US in 2030, and an increase in multimorbidity burden from 63% in 2020 to 70% in 2030 among people with HIV.

- Although mental health conditions had the highest burden, the mix of physical comorbidities of greatest burden was different by demographic and HIV acquisition risk subgroups.

### What do these findings mean?

- HIV clinicians can use these findings to identify the comorbidity-specific screening, diagnoses, and treatment guidelines and tools that will be necessary to care for their panel of patients.

- HIV care program decision-makers can use these findings to predict the subspecialities that will be in highest demand by their clinical population and make the connections to subspecialists (bringing them into the HIV clinic or by referral) needed meet the healthcare needs of people with HIV.

- HIV policy decision-makers can use these findings to guide expansion in subspecialty care capacity (particularly for mental health conditions) and the resources needed for expansion.

## Introduction

People with HIV (PWH) survive to older ages with effective antiretroviral treatment but have fewer comorbidity-free life years compared to people without HIV [1]. Forecasting the magnitude of mental and physical comorbidity, and multimorbidity, is critical for preparing to meet the future healthcare needs of people aging with HIV.

Many risk factors for comorbidities in the general population have a higher prevalence among people with HIV, including tobacco and substance use, higher body mass index (BMI), and hepatitis C virus (HCV) coinfection [2–5]. The increased risk profile contributes to a higher prevalence of major depressive disorder (depression), generalized anxiety disorder (anxiety), hypertension, dyslipidemia, chronic kidney disease (CKD), diabetes, liver disease, cancer, and cardiovascular disease (CVD) in people with (versus without) HIV [6–15]. Other factors contributing to the increased prevalence of comorbidities in people with HIV include: (1) HIV-induced chronic immune activation and inflammation [16,17]; (2) specific antiretroviral drugs and regimens [18,19]; and (3) social determinants of health (SDoH) [20,21]. Key SDoH include race, ethnicity, sex, and HIV acquisition risk group. The disproportionate prevalence of comorbidities and subsequent multimorbidity (i.e., ≥2 comorbidities not including HIV) pose persistent challenges in ensuring adequate healthcare for people with HIV [22,23]. Multimorbidity estimates among PWH in the US have ranged from 8.2% in 2000 to 65% in 2010 to 2011; the variability is in part due to modification to the definition of multimorbidity and the comorbidities included [24–26].

Disparities persist in the comorbidity and multimorbidity prevalence among PWH in the United States (US). People of color with HIV have greater comorbidity prevalence, which is accentuated in women of color [27,28]. Age-stratified incidence rates and risk of hypertension, diabetes, CKD, myocardial infarction (MI), and certain cancers are particularly high among non-Hispanic Black/African American (Black/AA) PWH, as are risk factors common to many comorbidities, including smoking, obesity, and SDoH [11,14,29,30]. People with injection drug use as their HIV acquisition risk factor have the greatest prevalence of comorbidity and multimorbidity among PWH [31,32]. To address healthcare inequities among subgroups of PWH, clinical program and policy decision-makers need forecasts of future multimorbidity burden within PWH subgroups [33,34]. The objective of this study is to forecast the prevalence of future comorbidities and multimorbidity among PWH using antiretroviral therapy (ART) in the US through the year 2030, overall and within 15 demographic subgroups.

## Methods

The ProjEcting Age, multimoRbidity, and poLypharmacy (PEARL) model is an agent-based computer simulation model of 15 subgroups of PWH who have initiated ART in the US, including those who disengage from HIV care (**Fig 1A**). The 15 subgroups are defined as: (1 and 2) men and women with history of injection drug use as an HIV acquisition risk factor, including those who have injection drug use and any additional HIV acquisition risk category specified (MWID and WWID, respectively); (3) men who have sex with men (MSM); and

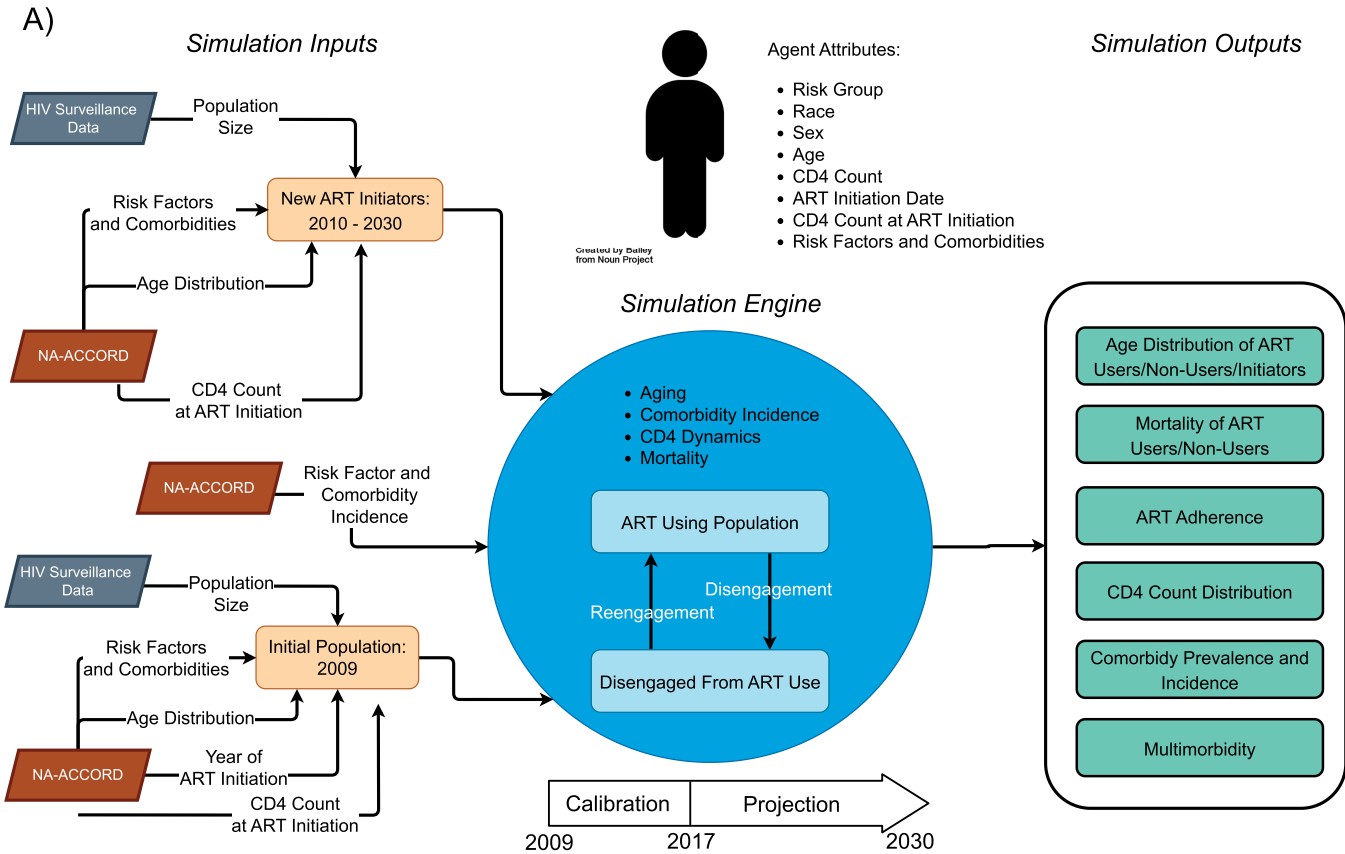

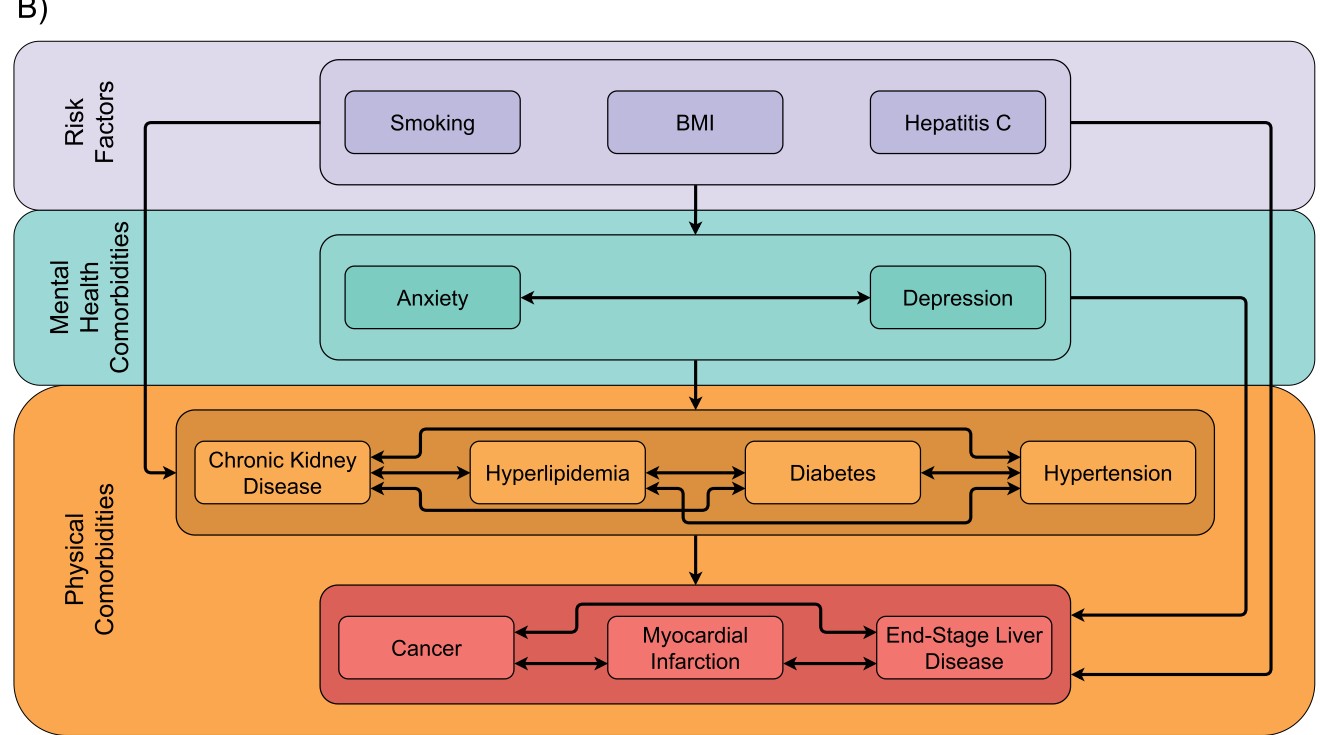

**Fig 1.** Schematic representation of (A) the PEARL model and (B) the risk factors and comorbidities with high prevalence in people with HIV using ART. (A) PEARL model simulating people with HIV using ART in the United States. Footnotes: HIV Surveillance data was sourced from the US Centers for Disease Control and Prevention's HIV Surveillance Reports, available at https://www.cdc.gov/hiv/library/reports/hiv-surveillance.html. The NA-ACCORD data was available after the collaboration approved our submitted concept sheet (https://naaccord.org/collaborate-with-us). (B) Schematic of the risk factors and comorbidities with high prevalence in people with HIV using ART. Footnotes: ART = antiretroviral therapy (HIV treatment). CD4 = CD4 T-lymphocyte cell count. Details on the mathematical functions represented by the arrows between the risk factors and mental and physical comorbidities can be found at PEARLHIVmodel.org.

(4 and 5) heterosexual men and women. These 5 groups were further stratified into non-Hispanic White, non-Hispanic Black/AA, and Hispanic. Race and ethnicity defined subgroups that compose the PEARL simulation model due to the disproportionate prevalence of HIV by race and ethnicity in the US. Asian and American Indian/Alaskan Native participants are not included in this analysis due to limited input parameters and functions within the 5 HIV acquisition risk groups.

Briefly, an initial population of agents with HIV using ART was constructed in 2009, and new agents are populated in their calendar year of ART initiation from 2010 to 2030; the characteristics of the agents reflect the observed characteristics of PWH initiating ART in the US informed by observed data from the North American AIDS Cohort Collaboration on Research and Design (NA-ACCORD) and Centers for Disease Control and Prevention's (CDC) HIV surveillance reports (**Fig 1A**) [35,36]. The agents are followed after ART initiation and observed to experience aging, disengagement, and re-engagement in care, changing CD4 counts, risk factors, comorbidities, and mortality via mathematical functions from the NA-ACCORD and CDC input parameters. The mathematical functions forecast the agents' experience in the future after observed data end. All mathematical functions and parameters in PEARL are estimated separately for each of the 15 subgroups or collapsed if a priori specifications of minimum sample size or number of events were not met (collapsing race/ethnicity groups first, followed by sex and then HIV-acquisition risk until the minimum is met). Due to the structure of PEARL and the data sources, findings are generalizable to PWH using ART in the US who identify with the 15 subgroups included in PEARL. Further methodological details regarding the model structure, parameterizations, standards for collapsing subgroups to ensure adequate sample size, and estimated functions are available at https://pearlhivmodel.org/method_details.html [35,36]. The code for the PEARL model results presented here can be found at https://github.com/PearlHivModelingTeam/comorbidityPaper.

## Comorbidities parameterization

First, we characterized the prevalence of clinical risk factors that are highly prevalent in PWH, linked to numerous comorbidities, and measured in the NA-ACCORD (namely smoking, obesity, and HCV coinfection) for the simulated population in 2009 and those starting ART over time. Smoking and HCV coinfection statuses were determined at ART initiation (i.e., the simulated person's entry to the model) and were time-fixed. Obesity (BMI $\geq 30$ kg/m$^2$) status was determined at ART initiation and at 24 months after ART initiation. Next, we estimated the prevalence (at ART initiation) and incidence (in the years after ART initiation) of highly prevalent comorbidities in PWH for simulated persons within the 15 subgroups via mathematical functions derived from observed NA-ACCORD data from 2009 to 2017; NA-ACCORD definitions for risk factors and comorbidities using electronic health record data are available in **S1 Table**. The prevalence of having depression, anxiety, stage $\geq 3$ CKD, dyslipidemia, type 2 diabetes, hypertension, cancer (all types), MI, and ESLD (at or prior to ART initiation) was estimated at the time the simulated person initiated ART (**Fig 1A**) [14,24,37–39]. Incidence of each comorbidity was estimated as a function of age, CD4 at ART initiation (cells/μL), time

since ART initiation, disengagement from care, change in BMI in the first 2 years after ART initiation, BMI 2 years after ART initiation, smoking, HCV coinfection, and the presence of the other comorbidities for simulated persons in the years after ART initiation (**S2 Table**). Finally, mortality was estimated for simulated persons as a function of individual-level attributes, existing risk factors, and present comorbidities, and was estimated separately for those (1) engaged and (2) disengaged from care ($\geq$2 years without CD4 or HIV RNA measurement) and in each of the 15 subgroups using observed NA-ACCORD data from 2009 to 2017 (**S3 Table**).

## Primary outcomes

While prior multimorbidity studies in PWH were restricted to physical comorbidities, we chose the following non-mutually exclusive definitions to produce findings comparable to prior studies and to expand the scope of multimorbidity to include the presence of mental health conditions:

a. physical multimorbidity ($\geq$2 physical comorbidities)

b. depression and/or anxiety diagnosis (mental comorbidities)

c. mental or physical multimorbidity ($\geq$2 mental or physical comorbidities), and

d. mental comorbidity and physical multimorbidity ($\geq$1 mental health comorbidity and $\geq$2 physical comorbidities).

Given the association of age with comorbidities of interest and heterogeneities in forecasted age distributions by subgroup, we report the forecasted median age in 2020 and 2030 within the 15 subgroups, as well as the forecasted prevalence of each mental and physical comorbidity, overall and within subgroups. To navigate temporal trends, we also report the absolute percentage point change (ppc) in the prevalence of each mental and physical comorbidity from 2020 to 2030 within the 15 subgroups.

Simulations were replicated for 200 runs and forecasted outcomes are presented as median values and the 95% uncertainty range (calculated as the 2.5th to 97.5th percentile of simulated values) in the Supporting information.

## Validation

We compared the estimated annual incidence of each comorbidity from the NA-ACCORD data with the forecasts from PEARL during the calibration period (where both observed NA-ACCORD estimates and PEARL forecasts were available, i.e., 2009 to 2017). Within each subgroup, we noted the comorbidities with <75% of PEARL forecasts falling within 5% of the NA-ACCORD observed incidence prevalence or within the NA-ACCORD observed 95% confidence (whichever interval was larger, **S1 Fig**). We repeated this validation approach for the annual prevalence of each comorbidity (**S2 Fig**).

## Robustness of forecasted comorbidity incidence

In the PEARL model, forecasted multimorbidity is a function of the comorbidities that arise from estimated probabilities for the incidence of each comorbidity within each subgroup. To assess the influence of the estimated probabilities for the incidence of each comorbidity on forecasted physical multimorbidity, we increased and decreased the probabilities for the incidence of each comorbidity by 25% for each simulated person in the model (i.e., the increase and decrease scenarios). We estimated the relative difference in the forecasted physical

multimorbidity in 2030 in each scenario compared to the baseline scenario (i.e., no change in the estimated probabilities for the incidence of each comorbidity). We chose physical multimorbidity as the outcome for the robustness checks to allow for comparability to other estimates of physical multimorbidity in PWH in the US [24–26]. The robustness of the prevalence and mortality estimates were similar, and results can be found at PEARLHIVMODEL.org/method_details.html.

### Ethics statement

The PEARL model was classified as "Exempt under 45 CRF 46.101(b), Category (4)" by the Johns Hopkins Bloomberg School of Public Health Institutional Review Board.

## Results

Using the PEARL model, we simulated a median population of 670,036 PWH using ART in 2020 in the US, of whom 52% were ≥50 years, 11% were age ≥65 years, 32% White, 44% Black/AA, 23% Hispanic, 60% MSM, 19% heterosexual women, 9% MWID, 8% heterosexual men, and 4% WWID, (**Tables 1** and **S4** for 95% UR). In 2020, the prevalence of anxiety and depression were 36% and 47%, respectively, with 23% of simulated agents having both diagnoses. Applying our non-mutually exclusive definitions of multimorbidity among ART users, 38% were physically multimorbid, 63% had mental or physical multimorbidity, and 25% had mental comorbidity and physical multimorbid. Of the physical conditions included in the model, dyslipidemia was the most prevalent (42%), followed by hypertension (37%), CKD (19%), diabetes (18%), and cancer (11%); MI and ESLD had a prevalence of <5%.

In 2030, the forecasted population of PWH using ART increased by almost a quarter of a million people (36% increase from 2020), and the proportion ≥65 years more than doubled to 23%; race/ethnicity and HIV acquisition risk group distributions changed by <5 percentage points from 2020 to 2030 (**Tables 1** and **S4** for 95% UR). The prevalence of hypertension decreased by 5 percentage points from 2020 to 2030, the prevalence of CKD and diabetes increased by 11 percentage points, diabetes increased by 9 percentage points, dyslipidemia increased by 6 percentage points, and MI increased by 5 percentage points; and the prevalence of cancer and ESLD remained constant.

The age distributions from NA-ACCORD participants and PEARL estimates were similar within subgroups from 2010 to 2017 (**S5 Table** and **S3 Fig**). Validity of the mental and physical comorbidity forecasts was confirmed by comparing the observed comorbidity incidence and prevalence of PWH using ART from 2010 to 2017 in NA-ACCORD with the simulated outcomes, suggesting no pattern of bias in forecasts from the model among 15 subgroups over time (**S1** and **S2 Figs**).

### Forecasts of multimorbidity

Overall, the prevalence of physical multimorbidity increased from 2020 to 2030 (**Fig 2A**) and in each subgroup (**Fig 2B**). The number of Black/AA MWID and WWID using ART decreased by 14% and 7% from 2020 and 2030 (respectively), and the physical multimorbidity prevalence was high and increased throughout the decade (**Tables 1 and S4** for 95% UR, **S6 Table**). Black/AA MSM and Black/AA heterosexual women had a 31% and 52% increase in the number of ART users from 2020 to 2030 (respectively). Physical comorbidity prevalence increased from 2020 to 2030 similarly to in Black/AA and White MSM, and the increase was greater in Black/AA compared with White or Hispanic heterosexual women.

Overall, the prevalence of depression and/or anxiety was higher than any physical comorbidity in 2020 (60%) and in 2030 (64%, **Fig 2A and S7 Table** for 95% UR). The prevalence

**Table 1. Characteristics of the PEARL-simulated agents using ART in 2010, 2020, and 2030.**

| | 2010 | | 2020 (Forecast) | | 2030 (Forecast) | |
|---|---|---|---|---|---|---|
| | PEARL[a] | | PEARL[a] | | PEARL[a] | |
| Characteristics | N = 395,062 | | N = 670,036 | | N = 908,504 | |
| | *n* | %[b] | *n* | %[b] | *n* | %[b] |
| **Age (in years)** | | | | | | |
| <20 | 694 | 0% | 708 | 0% | 1,218 | 0% |
| 20–24 | 8,444 | 2% | 11,520 | 2% | 12,416 | 1% |
| 25–29 | 20,035 | 5% | 40,475 | 6% | 40,927 | 5% |
| 30–34 | 30,853 | 8% | 66,704 | 10% | 78,305 | 9% |
| 35–39 | 47,911 | 12% | 62,720 | 9% | 101,328 | 11% |
| 40–44 | 67,598 | 17% | 60,845 | 9% | 102,122 | 11% |
| 45–49 | 78,947 | 20% | 79,388 | 12% | 85,320 | 9% |
| 50–54 | 66,281 | 17% | 97,822 | 15% | 81,110 | 9% |
| 55–59 | 42,435 | 11% | 100,416 | 15% | 94,472 | 10% |
| 60–64 | 20,636 | 5% | 76,357 | 11% | 102,878 | 11% |
| 65–69 | 7,834 | 2% | 44,442 | 7% | 94,677 | 10% |
| 70–74 | 2,476 | 1% | 19,669 | 3% | 64,950 | 7% |
| ≥75 | 911 | 0% | 8,912 | 1% | 48,378 | 5% |
| **Male sex at birth** | 290,010 | 73% | 515,560 | 77% | 711,684 | 78% |
| **Race** | | | | | | |
| White | 145,691 | 37% | 216,936 | 32% | 259,610 | 29% |
| Black/AA | 167,480 | 42% | 296,512 | 44% | 395,026 | 43% |
| Hispanic | 81,919 | 21% | 156,835 | 23% | 255,037 | 28% |
| **Subgroups, *n* %** <br> **median age [IQR]** | | | | | | |
| MSM | 210,640 | 53% | 404,087 | 60% | 584,158 | 64% |
| | 45 | [37, 51] | 48 | [35, 57] | 47 | [37, 61] |
| White MSM | 102,906 | 26% | 156,068 | 23% | 178,730 | 20% |
| | 47 | [41, 53] | 54 | [44, 60] | 59 | [46, 67] |
| Black/AA MSM | 63,110 | 16% | 143,910 | 21% | 218,548 | 24% |
| | 42 | [33, 49] | 41 | [32, 53] | 42 | [35, 56] |
| Hispanic MSM | 44,624 | 11% | 104,138 | 16% | 188,012 | 21% |
| | 41 | [35, 47] | 44 | [34, 52] | 43 | [36, 56] |
| Men who injected drugs (MWID)[c] | 49,868 | 13% | 59,597 | 9% | 66,022 | 7% |
| | 52 | [47, 57] | 58 | [49, 64] | 58 | [39, 69] |
| White MWID | 17,532 | 4% | 23,462 | 4% | 28,978 | 3% |
| | 50 | [44, 55] | 56 | [47, 62] | 57 | [40, 67] |
| Black/AA MWID | 20,544 | 5% | 21,177 | 3% | 18,184 | 2% |
| | 54 | [50, 58] | 61 | [55, 66] | 62 | [33, 72] |
| Hispanic MWID | 11,784 | 3% | 14,920 | 2% | 18,526 | 2% |
| | 52 | [45, 57] | 57 | [47, 64] | 56 | [41, 69] |
| Women who injected drugs (WWID) | 25,822 | 7% | 28,193 | 4% | 31,962 | 4% |
| | 49 | [43, 54] | 57 | [50, 63] | 62 | [51, 70] |
| White WWID | 7,463 | 2% | 9,506 | 1% | 13,274 | 1% |
| | 46 | [39, 51] | 53 | [44, 59] | 56 | [46, 65] |
| Black/AA WWID | 14,569 | 4% | 14,682 | 2% | 13,661 | 2% |
| | 50 | [45, 55] | 59 | [53, 64] | 66 | [58, 72] |
| Hispanic WWID | 3,791 | 1% | 4,050 | 1% | 5,088 | 1% |

*(Continued)*

**Table 1.** (Continued)

| Characteristics | 2010 PEARL[a] N = 395,062 | | 2020 (Forecast) PEARL[a] N = 670,036 | | 2030 (Forecast) PEARL[a] N = 908,504 | |
|---|---|---|---|---|---|---|
| | n | %[b] | n | %[b] | n | %[b] |
| | 49 | [44, 54] | 58 | [51, 63] | 62 | [52, 70] |
| Heterosexual men | 29,507 | 7% | 51,900 | 8% | 60,984 | 7% |
| | 47 | [40, 53] | 53 | [44, 60] | 58 | [47, 67] |
| White heterosexual men | 3,488 | 1% | 6,839 | 1% | 9,390 | 1% |
| | 49 | [43, 55] | 55 | [46, 62] | 61 | [48, 69] |
| Black/AA heterosexual men | 19,169 | 5% | 34,154 | 5% | 39,292 | 4% |
| | 47 | [41, 53] | 53 | [45, 60] | 58 | [46, 66] |
| Hispanic heterosexual men | 6,855 | 2% | 11,009 | 2% | 12,806 | 1% |
| | 44 | [37, 52] | 51 | [43, 60] | 57 | [49, 67] |
| Heterosexual women | 79,234 | 20% | 126,139 | 19% | 165,104 | 18% |
| | 44 | [36, 51] | 51 | [42, 58] | 56 | [47, 65] |
| White heterosexual women | 14,296 | 4% | 20,968 | 3% | 28,974 | 3% |
| | 45 | [38, 52] | 52 | [44, 59] | 59 | [50, 67] |
| Black/AA heterosexual women | 50,084 | 13% | 82,536 | 12% | 105,384 | 12% |
| | 43 | [36, 51] | 50 | [42, 58] | 56 | [47, 65] |
| Hispanic heterosexual women | 14,863 | 4% | 22,806 | 3% | 31,550 | 3% |
| | 43 | [36, 51] | 50 | [41, 59] | 55 | [42, 66] |
| **Mental comorbidities** | | | | | | |
| Anxiety | 94,982 | 24% | 243,968 | 36% | 425,498 | 47% |
| Depression | 157,566 | 40% | 314,996 | 47% | 442,003 | 49% |
| Anxiety and/or depression | 209,474 | 53% | 402,042 | 60% | 584,875 | 64% |
| **Physical comorbidities** | | | | | | |
| Stage ≥3 CKD | 40,336 | 10% | 126,360 | 19% | 273,896 | 30% |
| Dyslipidemia | 126,548 | 32% | 283,674 | 42% | 435,806 | 48% |
| Diabetes | 46,480 | 12% | 119,172 | 18% | 246,172 | 27% |
| Hypertension | 146,465 | 37% | 246,736 | 37% | 295,184 | 32% |
| Cancer | 37,634 | 10% | 74,712 | 11% | 101,700 | 11% |
| ESLD | 4,771 | 1% | 8,950 | 1% | 12,974 | 1% |
| MI | 6,217 | 2% | 22,064 | 3% | 73,666 | 8% |
| **Physical multimorbidity** | | | | | | |
| No physical comorbidities | 123,164 | 31% | 203,713 | 30% | 250,184 | 28% |
| 1 physical comorbidity | 163,010 | 41% | 210,150 | 31% | 247,158 | 27% |
| ≥2 physical comorbidities | 108,880 | 28% | 256,174 | 38% | 410,940 | 45% |
| **Mental and physical multimorbidity** | | | | | | |
| ≥2 mental or physical comorbidities | 208,708 | 53% | 421,313 | 63% | 631,834 | 70% |
| ≥1 mental and ≥2 physical comorbidities | 58,656 | 15% | 166,433 | 25% | 283,828 | 31% |
| **ART status** | | | | | | |
| PWH using ART | 395,062 | | 670,036 | | 908,504 | |
| ART initiators | 25,116 | | 33,054 | | 33,334 | |

(Continued)

**Table 1.** (Continued)

| Characteristics | 2010 PEARL[a] N = 395,062 | | 2020 (Forecast) PEARL[a] N = 670,036 | | 2030 (Forecast) PEARL[a] N = 908,504 | |
|---|---|---|---|---|---|---|
| | n | %[b] | n | %[b] | n | %[b] |
| Disengaged from ART use[d] | 42,332 | | 41,572 | | 33,186 | |

[a]Values represent the median for each simulated outcome across 200 random simulation replications (see **Supporting information S4 Table** for 2.5th and 97.5th percentile range presented as the 95% uncertainty range).

[b]Percentages in this table are calculated using the median numerator (n) and median denominator (N) from 200 replications of the PEARL model; for each characteristic, percentages will sum to 100%.

[c]MSM who also have MWID as their HIV acquisition risk factor were included in the MWID HIV acquisition risk group.

[d]PEARL forecasts of PWH using ART do not include 41,572 and 33,186 people who initiated ART but were disengaged from care and not using ART in 2020 and 2030 (respectively) and people of race and ethnicities other than non-Hispanic White, non-Hispanic Black/AA, and Hispanic.

AA, African American; ART, antiretroviral therapy for HIV treatment; IQR, interquartile range, estimated as the 25th and 75th percentile range of results from running the simulation 200 times; MSM, men who have sex with men; PEARL, Projecting Age, Multimorbidity, and Polypharmacy in Adults with HIV; PWH, people with HIV.

with depression and/or anxiety in 2030 was greatest in Hispanic MWID (87%) and Hispanic heterosexual women (86%). The increase in physical multimorbidity burden from 2020 to 2030 was greatest in Hispanic heterosexual men (19 ppc) and White heterosexual men (18 ppc). The proportion of PWH using ART with physical or mental multimorbidity was 63% in 2020 and 70% in 2030 (**Fig 2A and S4 Table** for 95% UR). The prevalence with mental comorbidities and physical multimorbidity increased from 25% in 2020 to 31% in 2030 (**Fig 2A and S4 Table** for 95% UR). To demonstrate the influence of risk factors—other than age alone—on forecasted multimorbidity, the multimorbidity prevalence by decade of age is shown in **S4 Fig,** which depicts an increase in multimorbidity prevalence in all age groups ≥50 years.

## Forecasts of each comorbidity

Among all PWH using ART from 2020 to 2030 forecasted by the PEARL model, depression had the highest prevalence over the 10-year period (49% in 2030), and anxiety increased from 36% in 2020 to 47% in 2030 (**Fig 3A and S7 Table** for 95% UR). From 2020 to 2030, hypertension prevalence declined slightly (<5 ppc), and dyslipidemia, diabetes, and CKD increased (>5 ppc); in 2030, prevalence for these 4 physical comorbidities ranged from 27% for diabetes to 48% for dyslipidemia in 2030. Cancer and ESLD had little change in prevalence; however, cancer had a higher prevalence in 2030 (11%) than ESLD (1%). In comparison, MI was forecasted to increase from 3% in 2020 to 8% in 2030.

There were differences in the forecasted change in comorbidity burdens from 2020 to 2030 across the 15 subgroups (**Fig 3B**). The subgroup with the oldest median age (66 years) in 2030 was Black/AA WWID, and Black MSM had the youngest median age in 2030 (42 years, **Table 1** and **Fig 2B**). Among Black/AA WWID, dyslipidemia, CKD, anxiety, hypertension, depression, and diabetes were the most prevalent comorbidities in 2030, and MI had lower but increasing prevalence from 2020 to 2030 (**Figs 2B** and **S3H**). Cancer and ESLD had little change in prevalence and declined slightly from 2020 to 2030. Among Black MSM, depression had the highest prevalence in 2030 followed by CKD (which increased rapidly from 2020 to 2030) and hypertension (which decreased from 2020 to 2030, **Figs 2B** and **S3B**). Dyslipidemia, diabetes, and anxiety had a prevalence >20% in 2030 all increased from 2020 to 2030; cancer and ESLD prevalence was low (<10%) and did not change. MI prevalence increased but was

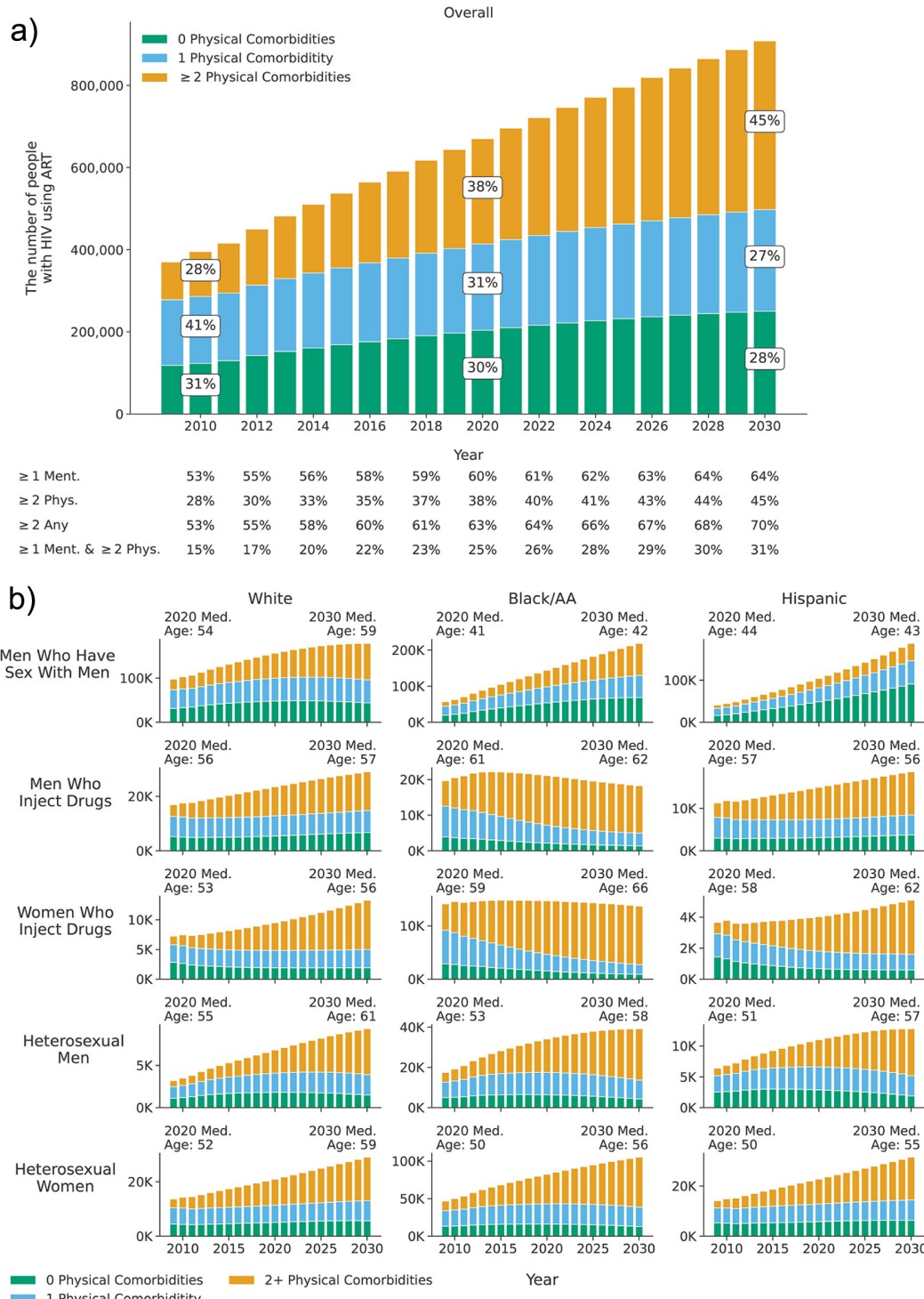

| | 2010 | 2012 | 2014 | 2016 | 2018 | 2020 | 2022 | 2024 | 2026 | 2028 | 2030 |
|---|---|---|---|---|---|---|---|---|---|---|---|
| ≥ 1 Ment. | 53% | 55% | 56% | 58% | 59% | 60% | 61% | 62% | 63% | 64% | 64% |
| ≥ 2 Phys. | 28% | 30% | 33% | 35% | 37% | 38% | 40% | 41% | 43% | 44% | 45% |
| ≥ 2 Any | 53% | 55% | 58% | 60% | 61% | 63% | 64% | 66% | 67% | 68% | 70% |
| ≥ 1 Ment. & ≥ 2 Phys. | 15% | 17% | 20% | 22% | 23% | 25% | 26% | 28% | 29% | 30% | 31% |

**Fig 2.** Forecasted[a] number of PWH using ART in the US and forecasted prevalence of mental and physical comorbidities and multimorbidity among PWH using ART in the US (A) overall and (B) by subgroup[b] (A) Overall. (B) By subgroup[b] Footnotes: ≥1 Ment. = anxiety and/or depression (i.e., ≥1 of the mental comorbidities included) ≥2 Phys. = physical multimorbidity, defined as ≥2 physical comorbidities ≥2 Any = mental or physical multimorbidity, defined as ≥2 physical or mental comorbidities ≥1 Ment. and 2 Phys. = mental comorbidity and physical multimorbidity, defined as ≥1 mental comorbidity and ≥2 physical comorbidities. [a]Although these estimates are all PEARL forecasts, 2010 was during the calibration period (where observed NA-ACCORD data were available to inform the estimates) and 2020 and 2030 were

forecast periods (without observed NA-ACCORD data). [b]Note that the y axes are different across the subgroups to allow visualization of the number of comorbidities within each year. ART, antiretroviral therapy; Black/AA, Black/African American; NA-ACCORD, North American AIDS Cohort Collaboration on Research and Design; PWH, people with HIV; US, United States.

<5% in 2030. Larger depictions of comorbidity prevalence estimates in each subgroup are available in **S5** **Figs**.

The ppc from 2020 to 2030 in the prevalence of each physical and mental comorbidity is shown in **Fig 4**, stratified by the 15 subgroups of PWH using ART. Overall, the prevalence of CKD, anxiety, diabetes, dyslipidemia, and MI increased by 11 ppc, 10 ppc, 9 ppc, 6 ppc, and 5 ppc (respectively). Dyslipidemia increased in all but 5 subgroups and by 6 ppc overall while hypertension decreased in all but 4 subgroups and by 4 ppc overall. There was <2 ppc change for depression, cancer, and ESLD.

## Robustness of each forecasted comorbidity incidence

The relative differences from analyzing the effect of decreasing or increasing the comorbidity incidence probability (versus baseline scenario) showed a <5% relative difference in the forecasted prevalence of physical multimorbidity, demonstrating the robustness of the forecasted multimorbidity to the variability in the estimated incidence of each comorbidity (**Fig 5**).

## Discussion

We forecast an increasing prevalence of multimorbidity in PWH using ART in the US through the year 2030, with different compositions of contributing comorbidities within race/ethnicity, gender, and HIV acquisition risk subgroups. Among the comorbidities included in the PEARL model, the forecasts suggest that 2 of the greatest contributors to multimorbidity are common mental health diagnoses that occur throughout the lifespan: depression and anxiety. The prevalence of anxiety was forecasted to increase in all the subgroups by the year 2030 and by 10 ppc overall. In 2030, the prevalence of mental or physical multimorbidity is forecasted to be 70% and nearly 1 in 3 (31%) PWH using ART will have mental comorbidity and physical multimorbidity by 2030; these estimates are conservative due to the inclusion of only 2 mental and 7 physical comorbidities when forecasting multimorbidity. Our findings show the most prevalent comorbidities in the next decade will differ by gender, HIV acquisition risk group, race, and ethnicity, suggesting the clinical population composition is important when preparing to meet the future care needs of people with HIV. HIV clinicians must be up-to-date on comorbidity-specific screening, diagnoses, and treatment guidelines and clinical decision-making tools that will be in highest demand by the subgroups represented in their panel of patients with HIV. HIV clinical directors must select comorbidities-specific refresher courses for their clinical staff and establish subspecialty care access within their clinics or by referral. HIV policy decision-makers must identify care models with capacity and ensure adequate payor resources (e.g., the Ryan White HIV/AIDS Program funding) to meet the growing healthcare needs, in particular the mental healthcare needs, of PWH using ART.

Our findings underscore the role of mental health comorbidities in PWH. PWH are 3 times more likely to currently be experiencing a major depressive episode as compared to people without HIV [40]. Not only has depression been linked to missed HIV clinical care visits, virologic failure, and all-cause mortality in PWH, but depression has also been linked to similar mechanisms of immune suppression and inflammation [41,42]. With the forecasted rise in anxiety prevalence among people with HIV, clinicians should consult the US Preventive Services Task Force's recommendation to screen for anxiety symptoms in those age <64 years

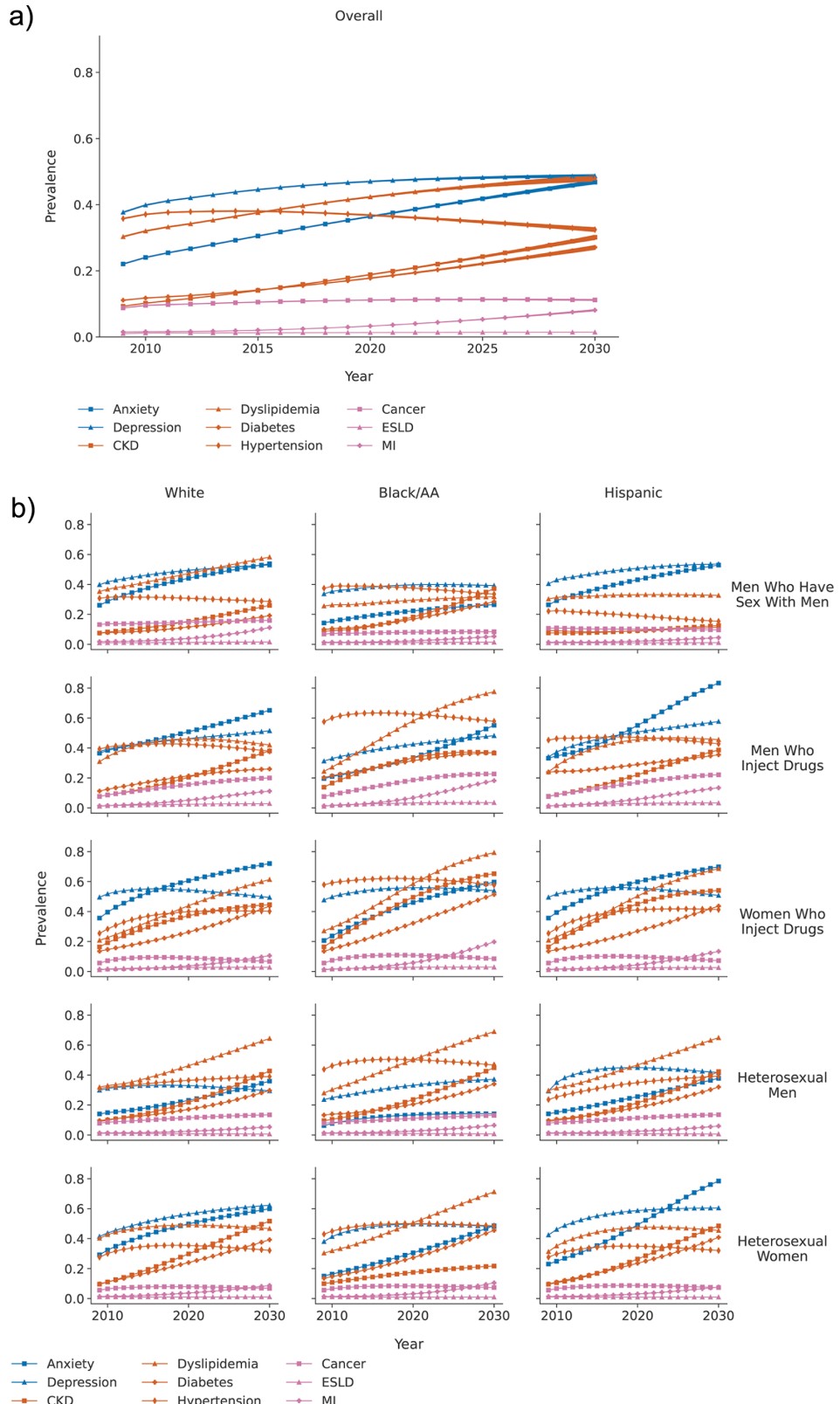

**Fig 3.** Forecasted prevalence (and shaded 95% uncertainty ranges) of individual comorbidities among PWH using ART (A) overall, (B) among the 15 subgroups, (C) among the subgroup with the oldest median age in 2030, and (C)

among the subgroup with the youngest median age in 2030. (A) Forecasted prevalence (and shaded 95% uncertainty ranges) of comorbidities among all PWH to the year 2030. (B) Forecasted prevalence (and shaded 95% uncertainty ranges) of individual comorbidities, within the 15 subgroups. Footnotes: CKD, stage ≥3 chronic kidney disease; ESLD, end-stage renal disease; MI, myocardial infarction. The 95% credibility interval is estimated as the 2.5% and 97.5% range of results from running the simulation 200 times.

(HIV-specific recommendations are not available), and be mindful of comorbidity medications that have been linked to symptoms of anxiety when caring for PWH [43]. Mental health services are allowable costs for PWH eligible for the federally allocated Ryan White HIV/AIDS Program support; funding mental health services within HIV clinics or by referral and ensuring adequate staffing will be necessary to meet the forecasted prevalence of mental health comorbidity.

Our findings corroborate estimates of increasing multimorbidity among PWH in the US and provide the opportunity to forecast future multimorbidity and comorbidity. A study comparing multimorbidity prevalence in PWH ages 45 to 89 years old attending 1 visit at a Ryan White HIV/AIDS Program clinic in 2016 versus 2006 observed an increase in multimorbidity prevalence among those of similar age and an increase in women (versus men) [44]. The PEARL-forecasted age-specific multimorbidity prevalence increased from 2020 to 2030 among those age 50 to 59, 60 to 69, and ≥70 years suggesting the contribution of risk factors—in addition to age itself—is resulting in an increased risk of comorbidities among older adults with HIV. A study of physical multimorbidity in the NA-ACCORD noted the common comorbidity composition included hypercholesterolemia, hypertension, and CKD in 2009 [24]. The prevalence of these metabolic and vascular diseases shown in the NA-ACCORD data

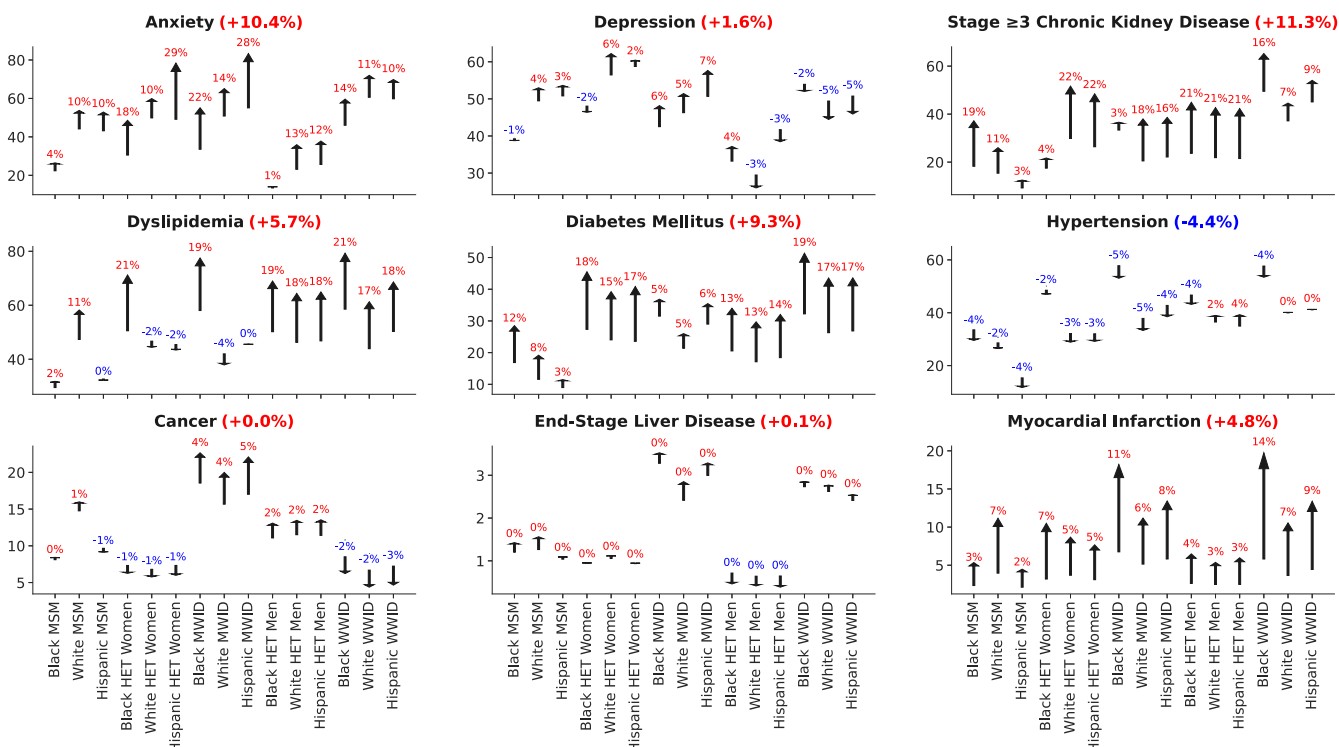

**Fig 4.** Forecasted absolute percentage point change (blue = decrease, red = increase) in the prevalence of individual comorbidities from 2020 to 2030, by subgroup. Footnotes: [b]The y axes are different across the subgroups to allow visualization of the number of comorbidities within each year.

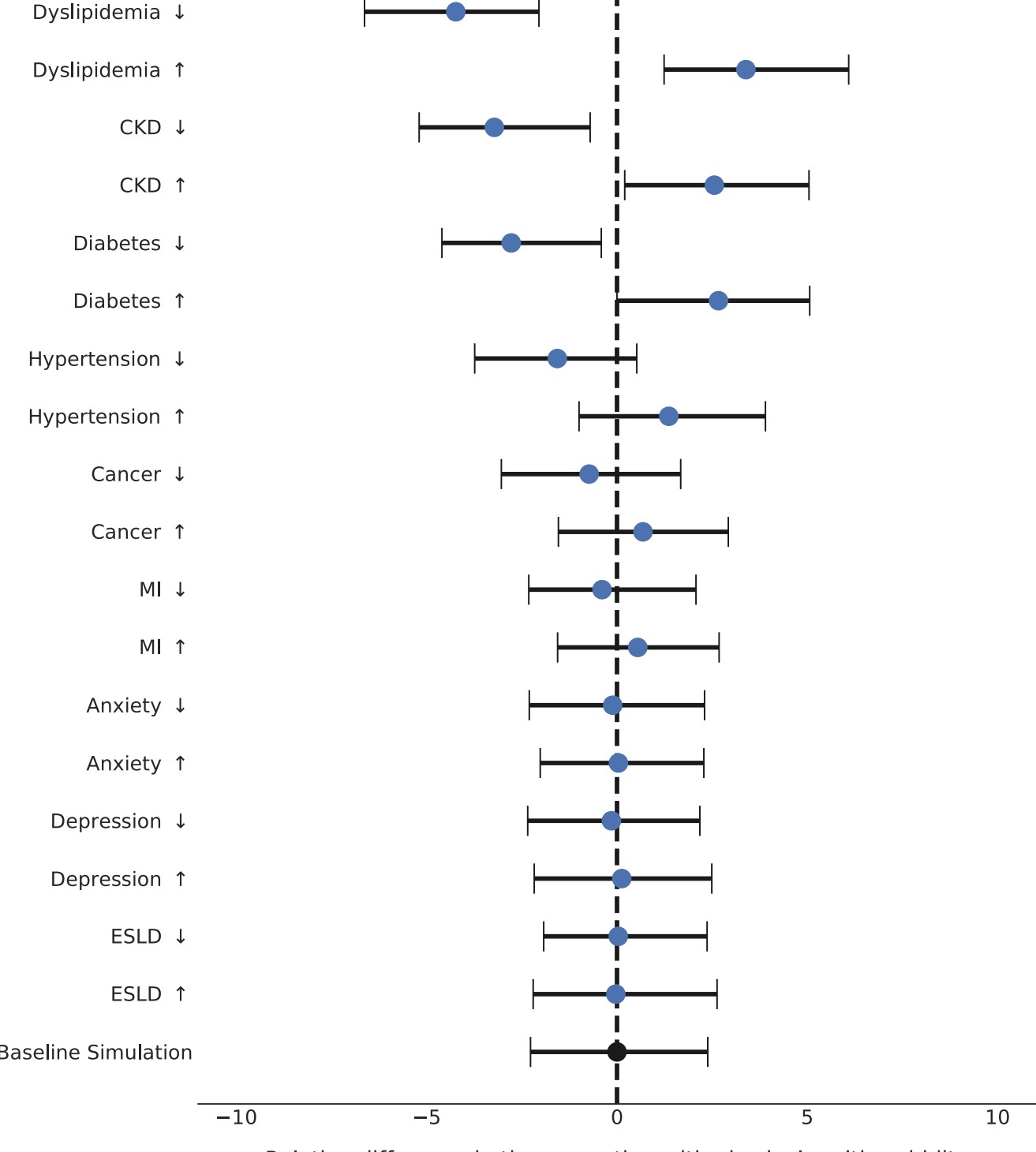

**Fig 5. The relative difference of the proportion with physical multimorbidity in 2030 [outcome] comparing scenarios in which comorbidity incidence was decreased by 25% (down arrow scenario) and increased by 25% (up arrow scenario) to assess the influence of estimated probabilities on prevalence estimates.** Relative difference when probability of the incidence of a comorbidity was decreased by 25% and increased by 25%, compared to the baseline

scenario (no modification to the probability of the incidence of a comorbidity). Footnotes: ↑relative difference = (% with physical multimorbidity $_{increase}$ $_{scenario}$—% with physical multimorbidity $_{baseline\ scenario}$) % with physical multimorbidity $_{baseline\ scenario}$ ↓relative difference = (% with physical multimorbidity $_{decrease\ scenario}$—% with physical multimorbidity $_{baseline\ scenario}$) % with physical multimorbidity $_{baseline\ scenario}$ Physical multimorbidity = ≥2 physical comorbidities.

influence the PEARL-forecasted increases in dyslipidemia, CKD, diabetes, and MIs. With the recent findings from the REPRIEVE Phase 3 clinical trial demonstrating a protective effect of pitavastatin on cardiovascular events, clinicians will need to stay informed on guideline changes for statins in people with HIV and recognize the gap between people with HIV-prescribed statins and those eligible for statins [45,46]. To forecast the potential impact of interventions to reduce the risk of future comorbidities, modules where a prominent risk factor is reduced by a specified amount are being added to the PEARL model.

Within each subgroup, the PEARL-estimated comorbidity combinations were influenced by the age distribution, physical and behavioral risk factors, and key SDoH that defined the subgroups (namely, race and ethnicity, gender, and HIV acquisition risk). Subgroup stratification is essential due to the heterogeneity of PWH in the US. For example, Black/AA WWID are forecasted to have the oldest age distribution in 2030 (median age = 66 years), which is a driven by trends in new HIV diagnosis (e.g., concentration of HIV diagnosis among middle ages over during 2009 to 2010 period, significant reduction in the number of new HIV diagnoses in all age groups over time) and HIV deaths (e.g., reduction in age-specific mortality rates among all age groups, concentration of deaths among older age groups), as shown in the CDC's HIV surveillance data from 2008 to 2021 [47]. Comparatively, the median age of White WWID is forecasted to reach 56 years by 2030, which is driven by the larger number of new HIV diagnoses (concentrated in young 25 to 44 years age groups) and a greater proportion of deaths at younger (35 to 54 years) ages [47]. A check of model calibration (for years 2010, 2013, and 2017) found the age distribution estimated by PEARL reflected the distribution observe in the NA-ACCORD within subgroups (**S3 Fig**) and the difference in these age distributions demonstrates the importance of stratification by subgroup. In addition to age differences, higher food insecurity, space for physical activity, and access to healthcare (influenced by structural racism and healthcare provider implicit bias/racism) can lead to higher rates of diabetes (forecasted to be 32% in Black/AA WWID in 2020) and hypertension (forecasted to be 62% prevalence in Black/AA WWID in 2020) which subsequently increases the risk of progression to renal failure, which is more prevalent in Black/AA (versus White) individuals in the US [48–50]. We forecasted CKD prevalence increased from 50% in 2020 to 65% in 2030 among Black/AA WWID. Through shared pathways of inflammatory mediators, reactive oxygen species (ROS), oxidative stress, and renin-angiotensin system (RAS) components, the 14 ppc increase in anxiety is also influencing the forecasted increase in CKD [51]. These comorbidities also influence the 14 ppc increase in MI among Black/AA WWID from 2020 to 2030. Implementing clinical program interventions that address the accessibility Black/AA women and focused on prevention and management of diabetes, hypertension, and anxiety may prove beneficial in slowing future multimorbidity growth among Black/AA WWID.

Our study has limitations. PEARL includes 9 highly prevalent comorbidities that necessitate clinical management, but it does not include arthritis, stroke, fractures, which were found to be among the top comorbidities noted in a recent UK analysis of 304 physical and mental health conditions in PWH; the PEARL-forecasted multimorbidity prevalence is likely an underestimate [52]. PEARL models clinically diagnosed conditions, which is beneficial for forecasting the needed clinical care resources but does not include undiagnosed conditions. PEARL model forecasts are currently at the national-level, and we are examining the

availability of data to forecast at the state-level. We did not compare the comorbidities and multimorbidity forecasts to similar people without HIV. Although this comparison would be useful, our goal was to provide future morbidity predictions to inform clinical planning and direct HIV policy decision-making. PEARL includes 15 subgroups defined by sex, HIV acquisition risk groups and race/ethnicity; however, it does not represent individuals who are multiracial or those with overlapping acquisition risks. ART regimen class is not explicitly contained in the PEARL model, but the impact of various ART regimen classes on comorbidity incidence is implicitly contained in the mathematical functions from observed NA-ACCORD data, which is reflective of exposures to ART in PWH in the US; future expansion of the PEARL model to include ART regimen class is possible. The PEARL model does not include HIV transmission. The forecasted annual number of new ART initiators relies on CDC's reported HIV diagnoses and linkage to care during the calibration period and (currently) does not consider changes in HIV transmission dynamics after the calibration period, including changes due to HIV prevention efforts (e.g., pre-exposure prophylaxis or PrEP) or during the COVID-19 pandemic (e.g., 2020 to 2021). Similarly, the calibration of mathematical functions in PEARL is based on NA-ACCORD data available during the calibration period. The PEARL model assumes that the trends observed during the calibration period will continue and does not account for influences occurring outside the calibration period (e.g., the impact of emerging care technologies like long-acting ART or the impact of COVID-19 on mortality); such factors influencing comorbidities incidence can be incorporated into PEARL by extending the calibration period when there are more current data available within the 15 subgroups.

As with all agent-based simulation models, the accuracy of the output is dependent upon the quality of the mathematical functions that compose the model. We utilized observed data from the NA-ACCORD to estimate the mathematical functions as it is the largest collaboration of PWH in the US and Canada and has similar demographics to all persons living with HIV in the US (according to the CDC's HIV surveillance data) [53]. Agent-based simulation models (such as PEARL) are recommended for chronic disease forecasting because they capture the complex interactions among individual-level risk factors that determine the risk of comorbidities and mortality as well as feedback loops needed for disengagement and re-engagement in care [54].

Multimorbidity is common in PWH using ART in the US and is likely to increase in prevalence over the next decade. Robust, sustainable, multidisciplinary care models (with appropriate funding) are urgently needed to meet the medically complex healthcare needs of PWH using ART in the US, in particular, access to affordable mental healthcare should be a priority. Predominant comorbidities differ by subgroups of PWH, which must be considered when planning for necessary resources and adapting care models. HIV clinicians must consider a host of comorbidity-specific guidelines to care for the increasing prevalence of comorbidities and multimorbidity among people with HIV. HIV clinical program and policy decision-makers must act now to identify effective multidisciplinary care models and resources to prevent and manage comorbidities and multimorbidity among the growing population of PWH using ART in the US.

## Supporting information

**S1 Fig. Comorbidity incidence validation plots, by subgroup.**
(DOCX)

**S2 Fig. Comorbidity prevalence validation plots, by subgroup.**
(DOCX)

**S3 Fig. Comparing the age distributions of ART users in PEARL to the observed data from NA-ACCORD, 2010, 2013, and 2017.**
(DOCX)

**S4 Fig.** Trends in multimorbidity distribution by age groups, overall and within the 15 subgroups of people with HIV, (a) overall; (b) White, (c) Black/African American, and (d) Hispanic men who have sex with men; (e) White, (f) Black/African American, and (g) Hispanic men with injection drug use as their HIV acquisition risk factor; (h) White, (i) Black/African American, and (j) Hispanic women with injection drug use as their HIV acquisition risk factor; (k) White, (l) Black/African American, and (m) Hispanic heterosexual men; (n) White, (o) Black/African American, and (p) Hispanic heterosexual women.
(DOCX)

**S5 Fig.** Forecasted prevalence (and shaded 95% credibility intervals) of individual comorbidities within the 15 subgroups of people with HIV: (a) White, (b) Black/African American, and (c) Hispanic men who have sex with men; (d) White, (e) Black/African American, and (f) Hispanic men with injection drug use as their HIV acquisition risk factor; (g) White, (h) Black/African American, and (i) Hispanic women with injection drug use as their HIV acquisition risk factor; (j) White, (k) Black/African American, and (l) Hispanic heterosexual men; (m) White, (n) Black/African American, and (o) Hispanic heterosexual women.
(DOCX)

**S6 Fig.** Ranges used to generate the number of new diagnoses, by HIV acquisition risk groups and race and ethnicity: (a) heterosexual women; (b) heterosexual men; (c) women who injected drugs; (d) men who injected drugs; (e) men who have sex with men.
(DOCX)

**S7 Fig.** Forecasted percentage of people linking to HIV care by HIV acquisition risk groups and race and ethnicity: (a) heterosexual females; (b) heterosexual males; (c) women who injected drugs; (d) men who injected drugs; (e) men who have sex with men.
(DOCX)

**S1 Table. Definitions of highly prevalent risk factors and comorbidities measured in the NA-ACCORD and included in the PEARL model.**
(DOCX)

**S2 Table. Prevalence and incidence functions applied to PEARL agents who have initiated ART for (a) anxiety prevalence, (b) anxiety incidence, (c) depression prevalence, (d) depression incidence, (e) stage ≥3 chronic kidney disease prevalence, (f) stage ≥3 chronic kidney disease incidence, (g) dyslipidemia prevalence, (h) dyslipidemia incidence, (i) diabetes prevalence, (j) diabetes incidence, (k) hypertension prevalence, (l) hypertension incidence, (m) cancer prevalence, (n) cancer incidence, (o) end-stage liver disease prevalence, (p) end-stage liver disease incidence, (q) myocardial infarction prevalence, and (r) myocardial infarction incidence.**
(DOCX)

**S3 Table. PEARL (a) in-care and (b) dis-engaged from care mortality functions that include comorbidity presence.**
(DOCX)

**S4 Table. Characteristics of the PEARL-simulated agents using ART, 2020 and 2030.**
(DOCX)

**S5 Table. Comparing the age distributions of ART users in PEARL to the observed data from NA-ACCORD, from 2010 to 2017 (simulation validation "out-of-sample" approach).** Values represent the difference in the age distribution in each subgroup [NA-ACCORD estimate—PEARL estimate]. A threshold of 5 percentage points (>5% or <-5%) is used to detect significant differences (highlighted in blue).
(DOCX)

**S6 Table. PEARL-forecasted multimorbidity prevalence, by year[a] and within each subgroup of PWH using ART in the US.**
(DOCX)

**S7 Table. PEARL-forecasted comorbidity and multimorbidity prevalence [95% uncertainty range], by subgroup, in 2010, 2020, and 2030.**
(DOCX)

**S8 Table. Number of new HIV diagnoses by year and subgroup.**
(DOCX)

## Acknowledgments

The mathematical functions upon which the PEARL model is built were estimated using observed data from the NA-ACCORD. The NA-ACCORD is supported by National Institutes of Health grants U01AI069918, F31AI124794, F31DA037788, G12MD007583, K01AI093197, K01AI131895, K23EY013707, K24AI065298, K24AI118591, K24DA000432, KL2TR000421, N01CP01004, N02CP055504, N02CP91027, P30AI027757, P30AI027763, P30AI027767, P30AI036219, P30AI050409, P30AI050410, P30AI094189, P30AI110527, P30MH62246, R01AA016893, R01DA011602, R01DA012568, R01AG053100, R24AI067039, R34DA045592, U01AA013566, U01AA020790, U01AI038855, U01AI038858, U01AI068634, U01AI068636, U01AI069432, U01AI069434, U01DA036297, U01DA036935, U10EY008057, U10EY008052, U10EY008067, U01HL146192, U01HL146193, U01HL146194, U01HL146201, U01HL146202, U01HL146203, U01HL146204, U01HL146205, U01HL146208, U01HL146240, U01HL146241, U01HL146242, U01HL146245, U01HL146333, U24AA020794, U54GM133807, UL1RR024131, UL1TR000004, UL1TR000083, UL1TR002378, Z01CP010214, and Z01CP010176; contracts CDC-200-2006-18797 and CDC-200-2015-63931 from the Centers for Disease Control and Prevention, USA; contract 90047713 from the Agency for Healthcare Research and Quality, USA; contract 90051652 from the Health Resources and Services Administration, USA; the Grady Health System; grants CBR-86906, CBR-94036, HCP-97105 and TGF-96118 from the Canadian Institutes of Health Research, Canada; Ontario Ministry of Health and Long Term Care, and the Government of Alberta, Canada. Additional support was provided by the National Institute Of Allergy And Infectious Diseases (NIAID), National Cancer Institute (NCI), National Heart, Lung, and Blood Institute (NHLBI), Eunice Kennedy Shriver National Institute Of Child Health & Human Development (NICHD), National Human Genome Research Institute (NHGRI), National Institute for Mental Health (NIMH) and National Institute on Drug Abuse (NIDA), National Institute On Aging (NIA), National Institute Of Dental & Craniofacial Research (NIDCR), National Institute Of Neurological Disorders And Stroke (NINDS), National Institute Of Nursing Research (NINR), National Institute on Alcohol Abuse and Alcoholism (NIAAA), National Institute on Deafness and Other Communication Disorders (NIDCD), and National Institute of Diabetes and Digestive and Kidney Diseases (NIDDK). Validated cancer data were collected by cancer registries participating in

the National Program of Cancer Registries (NPCR) of the Centers for Disease Control and Prevention (CDC).

The funders had no role in study design, data collection and analysis, decision to publish, or preparation of the manuscript. The content is solely the responsibility of the authors and does not necessarily represent the official views of the National Institutes of Health, or the US Center for Disease Control and Prevention, or Regeneron Pharmaceuticals Inc.

NA-ACCORD Collaborating Cohorts and Representatives.

AIDS Clinical Trials Group Longitudinal Linked Randomized Trials: Constance A. Benson and Ronald J. Bosch AIDS Link to the IntraVenous Experience: Gregory D. Kirk Emory-Grady HIV Clinical Cohort: Vincent Marconi and Jonathan Colasanti Fenway Health HIV Cohort: Kenneth H. Mayer and Chris Grasso HAART Observational Medical Evaluation and Research: Robert S. Hogg, Viviane Lima, Zabrina Brumme, Julio SG Montaner, Paul Sereda, Jason Trigg, and Kate Salters HIV Outpatient Study: Kate Buchacz and Jun Li HIV Research Network: Kelly A. Gebo and Richard D. Moore Johns Hopkins HIV Clinical Cohort: Richard D. MooreJohn T. Carey Special Immunology Unit Patient Care and Research Database, Case Western Reserve University: Jeffrey Jacobson Kaiser Permanente Mid-Atlantic States: Michael A. Horberg Kaiser Permanente Northern California: Michael J. Silverberg Longitudinal Study of Ocular Complications of AIDS: Jennifer E. Thorne MACS/WIHS Combined Cohort Study: Todd Brown, Phyllis Tien, and Gypsyamber D'Souza Maple Leaf Medical Clinic: Graham Smith, Mona Loutfy, and Meenakshi Gupta The McGill University Health Centre, Chronic Viral Illness Service Cohort: Marina B. Klein Multicenter Hemophilia Cohort Study–II: Charles Rabkin Ontario HIV Treatment Network Cohort Study: Abigail Kroch, Ann Burchell, Adrian Betts, and Joanne LindsayParkland UT Southwestern Cohort: Ank Nijhawan Retrovirus Research Center, Universidad Central del Caribe, Bayamon Puerto Rico: Angel M. Mayor Southern Alberta Clinic Cohort: M. John Gill and Raynell Lang Study of the Consequences of the Protease Inhibitor Era: Jeffrey N. Martin Study to Understand the Natural History of HIV/AIDS in the Era of Effective Therapy: Jun Li and John T. Brooks University of Alabama at Birmingham 1917 Clinic Cohort: Michael S. Saag, Michael J. Mugavero, and Greer Burkholder University of California at San Diego: Laura Bamford and Maile Karris University of North Carolina at Chapel Hill HIV Clinic Cohort: Joseph J. Eron and Sonia Napravnik University of Washington HIV Cohort: Mari M. Kitahata and Heidi M. Crane Vanderbilt Comprehensive Care Clinic HIV Cohort: Timothy R. Sterling, David Haas, Peter Rebeiro, and Megan Turner Veterans Aging Cohort Study: Kathleen McGinnis and Amy Justice.

NA-ACCORD Study Administration: Executive Committee: Richard D. Moore, Keri N. Althoff, Stephen J. Gange, Mari M. Kitahata, Jennifer S. Lee, Michael S. Saag, Michael A. Horberg, Marina B. Klein, Rosemary G. McKaig, and Aimee M. Freeman Administrative Core: Richard D. Moore, Keri N. Althoff, and Aimee M. Freeman Data Management Core: Mari M. Kitahata, Stephen E. Van Rompaey, Heidi M. Crane, Liz Morton, Justin McReynolds, and William B. Lober Epidemiology and Biostatistics Core: Stephen J. Gange, Jennifer S. Lee, Brenna Hogan, Elizabeth Humes, Sally Coburn, Lucas Gerace.

## Author Contributions

**Conceptualization:** Keri N. Althoff.

**Data curation:** Keri N. Althoff, Kelly Gebo, Amy C. Justice, Michael J. Silverberg, Michael A. Horberg, Viviane D. Lima, M. John Gill, Maile Karris, Peter F. Rebeiro, Jennifer Thorne, Heidi Crane, Mari Kitahata, Anna Rubtsova, Vincent C. Marconi, Gypsyamber D'Souza, Hyang Nina Kim, Sonia Napravnik, Kathleen McGinnis, Gregory D. Kirk, Timothy R. Sterling, Richard D. Moore.

**Formal analysis:** Cameron Stewart, Elizabeth Humes.

**Funding acquisition:** Keri N. Althoff, Cynthia Boyd, Kelly Gebo, Amy C. Justice.

**Investigation:** Keri N. Althoff, Cameron Stewart, Elizabeth Humes, Parastu Kasaie.

**Methodology:** Keri N. Althoff, Cameron Stewart, Elizabeth Humes, Emily P. Hyle, Sally B. Coburn, Raynell Lang, Peter F. Rebeiro, Ashleigh J. Rich, Cherise Wong, Sean Leng, Richard D. Moore, Parastu Kasaie.

**Project administration:** Keri N. Althoff, Lucas Gerace.

**Resources:** Keri N. Althoff.

**Supervision:** Keri N. Althoff.

**Validation:** Keri N. Althoff, Cameron Stewart.

**Visualization:** Keri N. Althoff, Cameron Stewart.

**Writing – original draft:** Keri N. Althoff.

**Writing – review & editing:** Keri N. Althoff, Cameron Stewart, Elizabeth Humes, Lucas Gerace, Cynthia Boyd, Kelly Gebo, Amy C. Justice, Emily P. Hyle, Sally B. Coburn, Raynell Lang, Michael J. Silverberg, Michael A. Horberg, Viviane D. Lima, M. John Gill, Maile Karris, Peter F. Rebeiro, Jennifer Thorne, Ashleigh J. Rich, Heidi Crane, Mari Kitahata, Anna Rubtsova, Cherise Wong, Sean Leng, Vincent C. Marconi, Gypsyamber D'Souza, Hyang Nina Kim, Sonia Napravnik, Kathleen McGinnis, Gregory D. Kirk, Timothy R. Sterling, Richard D. Moore, Parastu Kasaie.

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
