## [Editor Report · Decision Letter 0]

30 Dec 2022

Dear Dr Althoff, 

Thank you for submitting your manuscript entitled "The projected prevalence of comorbidities and multimorbidity in people with HIV in the United States through the year 2030" for consideration by PLOS Medicine.

Your manuscript has now been evaluated by the PLOS Medicine editorial staff as well as by an academic editor with relevant expertise and I am writing to let you know that we would like to send your submission out for external peer review.

Please re-submit your manuscript within two working days, i.e. by Jan 03 2023 11:59PM.

Kind regards,

Philippa Dodd, MBBS MRCP PhD

Senior Editor

PLOS Medicine

---

## [Decision Letter · Decision Letter 1]

30 May 2023

Dear Dr. Althoff,

Thank you very much for submitting your manuscript "The projected prevalence of comorbidities and multimorbidity in people with HIV in the United States through the year 2030" (PMEDICINE-D-22-03963R1) for consideration at PLOS Medicine. 

[LINK]

In light of these reviews, I am afraid that we will not be able to accept the manuscript for publication in the journal in its current form, but we would like to consider a revised version that addresses the reviewers' and editors' comments. Obviously we cannot make any decision about publication until we have seen the revised manuscript and your response, and we plan to seek re-review by one or more of the reviewers. 

We expect to receive your revised manuscript by Jun 20 2023 11:59PM. Please email us (plosmedicine@plos.org) if you have any questions or concerns.

We look forward to receiving your revised manuscript. 

Sincerely,

Philippa Dodd, MBBS MRCP PhD

PLOS Medicine

plosmedicine.org

GENERAL

Please respond to all editor and reviewer comments detailed below, in full.

Please include line and page numbers in your revised version.

COMMENTS FROM THE ACADEMIC EDITOR

I agree that this is potentially an interesting paper, for the US readers and policy-makers. It projects the potential burden of multimorbidity in people on ART by ethnic, sex and risk group, in the USA.

However, what I think is missing is the information that a clinician would like to have (rather than a policy/programme person) in understanding whether there is a greater or lesser likelihood that the person with HIV on ART under his/her care will develop multimorbidities, and which ones. With that knowledge some preventive action may be possible. Would it be possible to use the output from the model to give an example of what change there would be from 2020 to 2030 for such a clinician? The projected prevalences of multimorbidities are not that high, and although this is important to know about a population-level, it is not entirely clear for the clinician.

Also, the age distribution in PEARL 2020 and projected PEARL 2030 is somewhat different. How much of the increase is associated with changes in the projected age distribution of people on ART?

The model allows for age, but is it possible to clarify whether some (or all) of the projected multimorbidity increase is related to increases in particular age groups (as opposed to a general increase overall)? Would there be changes in the distribution of multi-morbidity by age in 2030, compared to 2020? What drives this increase?

Further, why is there no allowance for ART regimen? And ART history? Both regimen and previous history could bear on prevalence/incidence of morbidity. This is something that should at least be discussed, and if the data are not available then it should be noted under limitations.

It is not clear to me how specific to the PEARL model and available data this model is - can it be generalised across the USA (assuming not all HIV infected people start ART and likelihood of starting may change over time). This discussion could be informed by providing some understanding of what drives the increases in multimorbidity.

I agree with the insightful and pertinent comments from the reviewers, and consider all to indicate a major revision.

COMMENTS FROM THE EDITORS

COMPETING INTERESTS 

For those authors without competing interests for the purpose of brevity please list initials as opposed to full names.

FUNDING STATEMENT

Please remove the funding statement from page 2 and include only in the manuscript submission form. It will be compiled as metadata in the event of publication.

NA-ACCORD COLLABORATING COHORTS AND REPRESENTATIVES

Please move these details to an acknowledgement section at the end of the main manuscript

KEY POINTS

Please remove this subsection from the manuscript 

TITLE

Please revise your title according to PLOS Medicine's style. Your title must be nondeclarative and not a question. It should begin with main concept if possible. "Effect of" should be used only if causality can be inferred, i.e., for an RCT. Please place the study design ("A randomized controlled trial," "A retrospective study," "A modelling study," etc.) in the subtitle (ie, after a colon).

ABSTRACT

Please structure your abstract using the PLOS Medicine headings (Background, Methods and Findings, Conclusions).

Please combine the Methods and Findings sections into one section, “Methods and findings”.

Abstract Background: Provide the context of why the study is important. The final sentence should clearly state the study question.

Abstract Methods and Findings:

Please ensure that all numbers presented in the abstract are present and identical to numbers presented in the main manuscript text.

Please include the study design, population (including brief details of baseline demographics) and setting, number of participants, years during which the study took place, length of follow up, and main outcome measures in this section of the abstract.

Please quantify the main results with 95% CIs (or URs in this case) and p values. When reporting 95% CIs please separate upper and lower bounds with commas as opposed to hyphens which can be confused with reporting of negative values. When reporting p values please report as p<0.001 and when higher as p=0.002, for example. If not reporting p values then for the purpose of transparent data reporting please clearly state the reasons why not.

Please include any important dependent variables that are adjusted for in the analyses.

In the last sentence of the Abstract Methods and Findings section, please describe the main limitation(s) of the study's methodology.

Abstract Conclusions:

Please address the study implications without overreaching what can be concluded from the data; the phrase "In this study, we observed ..." may be useful.

Please interpret the study based on the results presented in the abstract, emphasizing what is new without overstating your conclusions.

Please avoid vague statements such as "these results have major implications for policy/clinical care". Mention only specific implications substantiated by the results.

Please avoid assertions of primacy ("We report for the first time....")

AUTHOR SUMMARY

At this stage, we ask that you include a short, non-technical Author Summary of your research to make findings accessible to a wide audience that includes both scientists and non-scientists. The authors summary should consist of 2-3 succinct bullet points under each of the following headings:

• Why Was This Study Done? Authors should reflect on what was known about the topic before the research was published and why the research was needed.

• What Did the Researchers Do and Find? Authors should briefly describe the study design that was used and the study’s major findings. Do include the headline numbers from the study, such as the sample size and key findings. 

• What Do These Findings Mean? Authors should reflect on the new knowledge generated by the research and the implications for practice, research, policy, or public health. Authors should also consider how the interpretation of the study’s findings may be affected by the study limitations. In the final bullet point of ‘What Do These Findings Mean?’, please describe the main limitations of the study in non-technical language.

The Author Summary should immediately follow the Abstract in your revised manuscript. This text is subject to editorial change and should be distinct from the scientific abstract. Please see our author guidelines for more information: https://journals.plos.org/plosmedicine/s/revising-your-manuscript#loc-author-summary

INTRODUCTION

Please address past research and explain the need for and potential importance of your study. Indicate whether your study is novel and how you determined that. If there has been a systematic review of the evidence related to your study (or you have conducted one), please refer to and reference that review and indicate whether it supports the need for your study.

METHODS and RESULTS

Please justify/clarify your decision to model predictions no later than 2030 and please consider extending your modelled predictions to a later time point i.e. why 10 year projections from 2020-2030 and not 15 year?

I could not find your ethics statement in your methods section, please include and please provide the name(s) of the institutional review board(s) that provided ethical approval.

Results paragraph 1 – please separate upper and lower bounds of URs with commas as opposed to hyphens to prevent confusion with reporting of negative values. 

Please ensure that percentages are quantified with numerators and denominators.

Please ensure that ‘ppc’ is defined at first use, I may have missed it apologies if so.

MODELLING STUDIES

Of all authors of modelling studies, we ask that the authors include the following items derived from Geoffrey P Garnett, Simon Cousens, Timothy B Hallett, Richard Steketee, Neff Walker. Mathematical models in the evaluation of health programmes. (2011) Lancet DOI:10.1016/S0140-6736(10)61505-X. We think you have included everything but please review the list below and amend as necessary:

• Please provide a diagram that shows the model structure, including how the disease natural history is represented, the process and determinants of disease acquisition, and how the putative intervention could affect the system.

• Please provide a complete list of model parameters, including clear and precise descriptions of [the meaning of each parameter, together with the values or ranges for each, with justification or the primary source cited, and important caveats about the use of these values noted].

• Please provide a clear statement about how the model was fitted to the data [including goodness-of-fit measure, the numerical algorithm used, which parameter varied, constraints imposed on parameter values, and starting conditions].

• For uncertainty analyses, please state the sources of uncertainties quantified and not quantified [can include parameter, data, and model structure].

• Please provide sensitivity analyses to identify which parameter values are most important in the model. Uncertainty estimates seek to derive a range of credible results on the basis of an exploration of the range of reasonable parameter values. The choice of method should be presented and justified.

• Please discuss the scientific rationale for this choice of model structure and identify points where this choice could influence conclusions drawn. Please also describe the strength of the scientific basis underlying the key model assumptions.

FIGURES

Throughout, please consider avoiding the use of green and/or red to make your figures more accessible to those with colour blindness.

Throughout, please indicate whether your analyses are adjusted or unadjusted and where adjusted analyses are presented please detail the factors which are adjusted for in an appropriate caption/footnote and, to help facilitate transparent data reporting, please provide the unadjusted analyses for comparison.

Figure 1 – please see reviewer comments below which we agree with, please revise you figure accordingly.

Figure 2 – mental and physical are abbreviated to ‘ment.’ and ‘phys.’ Do they need to be? Suggest revising to include the complete words. 

In part a) you present percentages and in part b) you present total numbers of affected individuals – is there a reason for this difference, please clarify. We prefer the latter for both parts for improved clarity and consistency in reporting.

In part b) what does ‘Med’ mean at the top of each graph? Please clarify and define for reader in the footnote.

Figure 3 – is there a reason why the same colour scheme is used to define different comorbid illnesses? It is very difficult to discern which line is representative of what, especially in the smaller graphs in part b), please revise.

In the figure caption you refer to part c) but I was unable to locate a part c) at least in my version. Please clarify/revise.

Figure 5 – there is an unchecked tracked comment here, please remove. In a caption or footnote please clearly define the meaning of the dots and lines for the reader.

DISCUSSION

Please ensure that you present and organize the Discussion as follows: a short, clear summary of the article's findings; what the study adds to existing research and where and why the results may differ from previous research; strengths and limitations of the study; implications and next steps for research, clinical practice, and/or public policy; one-paragraph conclusion.

SUPPORTING INFORMATION

Table S1 – first column, final row – End-stage renal disease doesn’t seem to match appropriately to the abbreviation ‘(ESLD)’ should it read End-stage liver disease? Please clarify/revise accordingly.

Table S2 – please define the units of measurement for the prevalence data reported from the NA ACCORD.

Table S3 – is very small and rather inaccessible to the reader, please revise

Table S5 - Please present numerators and denominators for percentages

Figure S1 – suggest changing the angle of the labels on the axis ‘2010’ and ‘2015’ to a diagonal position to improve reader accessibility 

Figure S3a-o) – as for the main manuscript – please revise the choice of colour scheme to improve clarity for the reader.

REFERENCES

For in-text reference callouts please place citations in square brackets and preceding punctuation for example [1,2,3-6]. Please check and amend throughout including the supporting information where relevant.

In the bibliography, please ensure that web references detail an access date.

Journal name abbreviations should be those found in the National Center for Biotechnology Information (NCBI) databases. 

Please ensure that up to but no more than 6 author names are detailed followed by et al.

Please see our website for other reference guidelines https://journals.plos.org/plosmedicine/s/submission-guidelines#loc-references

Comments from the reviewers:

Reviewer #1: Thanks for sharing your manuscript entitled: "The projected prevalence of comorbidities and multimorbidity in people with HIV in the United States through the year 2030". I really enjoy reading your work, I think this kind of modeling efforts are very important to define the potential burden of diseases in a population, and to plan the care and needs of HIV people further than their HIV traditional care, even more if they are focused on groups of specific and different risk. 

I have only minor comments related to the description of the model. I couldn´t understand in the manuscript, how new people incorporates in the follow-up, and neither I could open the link which authors mention that details are (Please double-check the link is really working). I would like to see some lines into the manuscript to understand how is the entrance in the model by group of risk year by year. If this is explained in previous papers, I think readers interested could be informed. I would also like to know why ART regimen was not incorporated in some of the comorbidities´ incidence estimations? Is not a current potential association of ART with some of the variables involved, such as changes in BMI? Each risk group has their own probabilities of ART disengagement and re-engagement? 

Additionally, I´m curious about if this model could be used to predict the comorbidities by group by US state, Is this possible? Or the model calibration and other details to program it don´t allow it? I think this kind a state stratification could be useful for local policy makers and that other social factors associated with local resources, education level or income could be incorporated in the estimations. Is this an issue that authors could comment/discuss into their work? New potential policies to have a better care for some of the populations: those avoiding racism in black and Hispanic population, or those addressed on drug users, could have any impact on your estimations? How big could be this impact? 

I think authors could also mention some advantages of the agent-based models compared to other type of models in the discussion. 

Reviewer #2: This very interesting study aims to predict the rates of co-morbidities in different groups of US PLHIV.

Abstract and title are confusing because they do not indicate that this is a modeling study.

The Introduction discusses the criticality of SDoH, but as-written, it is not clear which SDoH are included in the model. Please incorporate these into Figure 1. For broader introduction, please also define SDoH and list which are the most important determinants for the populations modeled.

The last sentence of paragraph 2 of the Intro is difficult to parse: "Multimorbidity is a function of the comorbidities included in the definition and estimates among PWH in the US have a wide range." There seem to be two different points being made here. Make them separately and in sufficient detail, i.e., one sentence explaining the definition of multimorbidity, and the other stating the ranges on multimorbidity estimates from prior studies.

Is it necessary to subdivide factors correlated with but not caused by HIV (such as smoking an injecting drugs), versus ones caused by HIV (cancer and diseases associated with chronic immune activation)? 

The model appears to really be fifteen different models for each of 5 HIV transmission group (MSM, MWID, WWID, heterosexual men, heterosexual women) X 3 racial groups (white, black/AA, Hispanic). It does not consider that individuals can be multiracial or have overlapping risks. Does it include transmission between racial groups an risk groups? It is important to clarify these limitations and think about whether the results are still valid subject to these limitations.

The model appears to only include people from the time they first initiate ART. How is the rate of people newly initiating ART (i.e., new individuals entering the model) determined? Presumably this depends on future ART coverage as well as future rates of new HIV infections which give rise to people never before on ART - and HIV infections in turn depend on the populations with unsuppressed HIV viral load participating in different forms of behavioral risk - how is all of this handled?

Drug use and risk of drug overdose is a large driver of morbidity and mortality, in PWID, constituting six of the fifteen populations being modeled. How can the model accurately capture life expectancy without accounting for drug use morbidities and risk of overdose?

The calibration plots in Supplementary Appendix are an excellent component of this study and very well-presented. There don't appear to be any incidence plots where the model fails to match the data, but there are a number of prevalence plots where the model fails to match the data. How can this be? Is it a problem with the initial conditions?

The model estimates the incidence of comorbidities (e.g., depression) as a function of calendar year, age, CD4 count at ART initiation, and time since ART initiation. These are shown as betas from a regression. Does that assume that incidence of comorbidities must always be a linear function of age? Many comorbidities have an age of maximum or minimum incidence (hump-shaped of bowl-shaped function of age) and many others have low incidence until older age (aging-related conditions like HTN, diabetes…) How are these common age patterns taken into account?

Why did Black WWID have the oldest median age in 2030? One would think their life expectancy would be shorter than heterosexual women due to many health risks and comorbidities. Could this possibly be an error in the model? If so, this could explain why WWID are predicted to have so many co-morbidities compared to other groups (they are simply older and risks of many chronic health conditions grow with age).

Could the authors provide calibration graphs showing how their modeled population age structure compares to observed age structure of the 15 groups in different years? I wonder if discrepancies there are contributing to the unintuitive median age findings in the model outputs.

As a benchmark, could the authors show what co-morbidities the model would predict in HIV-negative people, and whether the model does a reasonable job of those predictions? This could help validate the comorbidity predictions in a broader population and troubleshoot issues like linear relationships with age.

Overall, a very interesting study, but the model may require further debugging and validation to ensure the reported predictions are accurate.

Reviewer #3: See attachment

Michael Dewey

[LINK]

---

## [Decision Letter · Decision Letter 2]

15 Sep 2023

Dear Dr. Althoff,

Thank you very much for submitting your manuscript "The forecasted prevalence of comorbidities and multimorbidity in people with HIV in the United States through the year 2030: A modeling study" (PMEDICINE-D-22-03963R2) for consideration at PLOS Medicine. 

[LINK]

In light of these reviews, I am afraid that we will not be able to accept the manuscript for publication in the journal in its current form, but we would like to consider a revised version that addresses the reviewers' and editors' comments. Obviously we cannot make any decision about publication until we have seen the revised manuscript and your response, and we plan to seek re-review by one or more of the reviewers. 

We expect to receive your revised manuscript by Oct 06 2023 11:59PM. Please email us (plosmedicine@plos.org) if you have any questions or concerns.

We look forward to receiving your revised manuscript. 

Sincerely,

Philippa Dodd, MBBS MRCP PhD

PLOS Medicine

plosmedicine.org

COMMENTS FROM THE EDITORS

GENERAL

The reviewer and the academic editor both raise serious concerns about the validity of your model which we agree with. Please see below. The reviewer has concerns that your response to previous comments is incomplete and we agree that the unexpected results should be investigated further. We cannot progress your manuscript without the aforementioned.

For this reason, we have invited a further major revision of the manuscript. 

Please address the comments detailed below in full.

FIGURES

Please consider avoiding the use of red and or green to improve accessibility of your figures to those with colour blindness.

Figure 5 - you clearly define the meaning of the dots and the lines for the reader in the caption.

DISCUSSION

Please remove the sub-heading ‘conclusion’ such that it reads as continuous prose.

REFERENCES

In the bibliography please list up to but no more than 6 author names followed by et al in the event than more than 6 authors contribute to an individual study.

Please ensure all web references include an ‘Accessed [date]’

Please see our website for other reference guidelines https://journals.plos.org/plosmedicine/s/submission-guidelines#loc-references

SUPPORTING INFORMATION

In the published article, supporting information files are accessed only through a hyperlink attached to the captions. For this reason, you must list captions at the end of your manuscript file. You may include a caption within the supporting information file itself, as long as that caption is also provided in the manuscript file. Do not submit a separate caption file.

Supplementary table 6 – in the footnote, please revised CDK to read CKD

COMMENTS FROM THE ACADEMIC EDITOR

I am a little unsure about this paper - the model is becoming more complex but the value of the paper for the readership of PLoS Med has become more, rather than less. And I do not see what is new knowledge?

I continue to struggle with who this paper is for - given the various caveats noted by the authors in their responses to the previous comments. I remain unclear as to how much of the increase in multimorbidity is due to the ageing of the HIV cohort in the USA, and whether there is any much change in the incidence/prevalence of multimorbidity by age forecast over the period to 2030.

Reviewer 2 has a point regarding the age of the black women with injecting drug use - but that may be because there is ageing of the USA cohort with younger cohorts having a different HIV acquisition factor compared to older cohorts. And also, in their population, injecting drug use only account for a small percentage of all acquisition risk.

To me, this paper does not show an understanding of HIV itself, and how HIV would interact with ageing. Further, the point about not being able to, or not seeing the need, to include ART regimen as a factor in the forecast model is not a satisfactory response. There may not be much data for individual ART regimen, but it should be possible to divide ART regimen in classes, on the basis of how they work, what they address and what side effects there may be. Surely, an HIV clinical expert would know this. To me this is an important point as there are changes in ART regimen and if those would be associated with changes in specific multimorbidity patterns than that is something policymakers and programme leaders should be aware of.

The conclusion of the abstract states:

Conclusion and relevance: The distribution of multimorbidity will continue to differ by race/ethnicity, gender, and HIV acquisition risk subgroups, and be influenced by age and risk factor distributions that reflect the impact of social disparities of the health on women, people of color, and people who use drugs. HIV clinical care models and funding are urgently required to meet the healthcare needs of people with HIV in the next decade.

But no specifics are provided and this leaves the reader wondering what then the paper contributes to knowledge useful for clinical care?

COMMENTS FROM THE REVIEWERS:

Reviewer #2: The authors have been responsive to most of the suggestions from this reviewer, except for two crucial ones:

- Longer LE in Black WWID: The authors postulate that Black women who inject drugs may have longer life expectancy due to gender differences in life expectancy, a survivor bias effect, or national reductions in HIV incidence among PWID. To me, none of these putative mechanisms make sense as a plausible driver of the model results - and some explanations simply do not make sense in the context of this model. I urge the authors to think deeply and investigate thoroughly, i.e., use their data and model to interrogate the underlying causes of this unexpected result. Spurious results are often useful "leads' that help to diagnose bugs in the model. Better to find out now than have this lead to a retraction of the article in the future.

- Fitting: The reviewer requested that the authors graph the age distributions from their modeled population vs. the observed population. Instead, the reviewer provided 8 pages of tables listing % differences between model and data in different strata. The requested graph would show the actual population by age group, graphed as a distribution, and overlaying these for model vs. data on the same graph. This visualization approach will be much more interpretable and revealing of the model's validity, versus the 8-page table.

Reviewer #3: The authors have addressed my points.

Michael Dewey

[LINK]

---

## [Decision Letter · Decision Letter 3]

7 Nov 2023

Dear Dr. Althoff,

Thank you very much for re-submitting your manuscript "The forecasted prevalence of comorbidities and multimorbidity in people with HIV in the United States through the year 2030: A modeling study" (PMEDICINE-D-22-03963R3) for review by PLOS Medicine.

I have discussed the paper with my colleagues and the academic editor and it was also seen again by one of the reviewers. I am pleased to say that provided the remaining editorial and production issues are dealt with we are planning to accept the paper for publication in the journal.

[LINK]

We look forward to receiving the revised manuscript by Nov 14 2023 11:59PM.   

Sincerely,

Philippa Dodd, MBBS MRCP PhD

PLOS Medicine

plosmedicine.org

pdodd@plos.org

COMMENTS FROM THE ACADEMIC EDITOR

The authors have done a good job in taking on board the comments from the reviewer and myself. I would be happy with your decision to proceed.

This paper is still a modelling paper but the authors now highlight where the results can be used in clinical care. They also note that they would be happy in further work to explore the impact of ART regimen on multimorbidity - which would lead to informing understanding as what is happening (rather than describing what is happening which is what the current paper does).

COMMENTS FROM THE EDITORS

Thank you for your detailed responses to previous reviewer and academic editor comments.

We did note that some of the previous editorial requests have yet to be incorporated into your revision, such as amendments to the abstract, for example. I have detailed these and other editorial requests below. 

Please also include a response (as for the reviewers) detailing your response and signposting to where the amendments can be located in the manuscript.

Please respond to all comments in full these are a requirement for publication.

DATA AVAILABILITY STATEMENT

In the manuscript submission form, please revise your statement to include all URLs pertaining to your data sources and model as detailed in your main manuscript. Suggest the following:

HIV Surveillance data was sourced from the US Centers for Disease Control and Prevention’s HIV Surveillance Reports, available at https://www.cdc.gov/hiv/library/reports/hiv-surveillance.html. 

NA-ACCORD data is available following approval from the collaboration https://naaccord.org/collaborate-with-us. 

Methodological details regarding the model structure, parameterizations, standards for collapsing subgroups to ensure adequate sample size, and estimated functions are available at https://pearlhivmodel.org/method_details.html. 

The code for the PEARL model results presented here can be found at https://github.com/PearlHivModelingTeam/comorbidityPaper.

COMPETING INTERESTS 

Please remove the competing interest statement from the main manuscript and include only in the manuscript submission form. It will be compiled as metadata at the time of publication.

FUNDING STATEMENT

Please remove the funding statement from page 2 and include only in the manuscript submission form. It will be compiled as metadata at the time of publication.

KEY POINTS

Please remove this subsection from the manuscript 

ABSTRACT

Please structure your abstract using the PLOS Medicine headings (Background, Methods and Findings, Conclusions).

Please combine the Methods and Findings sections into one section, “Methods and findings”.

Abstract Background: Provide the context of why the study is important. The final sentence should clearly state the study question.

Abstract Methods and Findings:

Please ensure that all numbers presented in the abstract are present and identical to numbers presented in the main manuscript text.

Please give further (brief) details of the database upon which you based your simulations – NA-ACCROD and CDC.

As in the current version, please include the study design, population (including brief details of baseline demographics) and setting, number of participants, years during which the study took place, length of follow up, and main outcome measures in this section of the abstract.

Please quantify the main results with 95% CIs (or URs in this case) and p values. When reporting 95% CIs please separate upper and lower bounds with commas as opposed to hyphens which can be confused with reporting of negative values. When reporting p values please report as p<0.001 and when higher as p=0.002, for example. If not reporting p values then for the purpose of transparent data reporting please clearly state the reasons why not.

Please include any important dependent variables that are adjusted for in the analyses.

In the last sentence of the Abstract Methods and Findings section, please describe the main limitation(s) of the study's methodology.

Abstract Conclusions:

Please address the study implications without overreaching what can be concluded from the data; the phrase "In this study, we observed ..." may be useful.

Please interpret the study based on the results presented in the abstract, emphasizing what is new without overstating your conclusions.

Please avoid vague statements such as "these results have major implications for policy/clinical care". Mention only specific implications substantiated by the results.

Please avoid assertions of primacy ("We report for the first time....")

AUTHOR SUMMARY

At this stage, we ask that you include a short, non-technical Author Summary of your research to make findings accessible to a wide audience that includes both scientists and non-scientists. The authors summary should consist of 2-3 succinct bullet points under each of the following headings:

• Why Was This Study Done? Authors should reflect on what was known about the topic before the research was published and why the research was needed.

• What Did the Researchers Do and Find? Authors should briefly describe the study design that was used and the study’s major findings. Do include the headline numbers from the study, such as the sample size and key findings. 

• What Do These Findings Mean? Authors should reflect on the new knowledge generated by the research and the implications for practice, research, policy, or public health. Authors should also consider how the interpretation of the study’s findings may be affected by the study limitations. In the final bullet point of ‘What Do These Findings Mean?’, please describe the main limitations of the study in non-technical language.

The Author Summary should immediately follow the Abstract in your revised manuscript. This text is subject to editorial change and should be distinct from the scientific abstract. Please see our author guidelines for more information: https://journals.plos.org/plosmedicine/s/revising-your-manuscript#loc-author-summary

FIGURES

Figure 5 – there is an unchecked tracked comment here, please remove.

DISCUSSION

We agree with the reviewer (please see below) that the 2030 projection showing black women who inject drugs being older than any other group in the model is an unusual finding that warrants further discussion. Please include.

REFERENCES

For in-text reference callouts please place citations in square brackets and preceding punctuation for example [1,3-6]. Please check and amend throughout including the supporting information where relevant.

SUPPORTING INFORMATION

Please ensure reference formatting follows the same guidance as for the main manuscript. For reference these are as detailed below:

In the bibliography, please ensure that web references detail an access date.

Journal name abbreviations should be those found in the National Center for Biotechnology Information (NCBI) databases.

Please ensure that up to but no more than 6 author names are detailed followed by et al.

Please see our website for other reference guidelines https://journals.plos.org/plosmedicine/s/submission-guidelines#loc-references

In the published article, supporting information files are accessed only through a hyperlink attached to the captions. For this reason, you must list captions at the end of your manuscript file. You may include a caption within the supporting information file itself, as long as that caption is also provided in the manuscript file. Do not submit a separate caption file.

I could only see supplementary file labels (e.g. Table S1) included – please also include a title and caption for each as outlined above (these were present in the previous version but perhaps not updated following revision).

SOCIAL MEDIA

To help us extend the reach of your research, please detail any X (formerly Twitter) handles you wish to be included when we tweet this paper (including your own, your co-authors’, your institution, funder, or lab) in the manuscript submission form when you re-submit the manuscript.

Comments from Reviewers:

Reviewer #2: I appreciate the authors' efforts to show their modeled age distributions in 2010, 2013, and 2017 as compared to in-care cohort data. It fits very well! I am still struggling to understand why the 2030 projection shows black women who inject drugs becoming an entire decade older than white women who inject drugs - and indeed older than any other group in the model. When I scan down the columns of 2010, 2013, and 2017 in Figure S3, the curves look similar by racial group for women who inject drugs, e.g., a 2017 peak of 50 years old. With the greyscale graph of deaths, the authors revealed in their response that deaths among Black women who inject drugs shifted to older age groups, suggesting this isn't a survival bias where the younger women have died and left only the older women alive, as the authors insinuated earlier. In summary, while I personally still struggle to get my head around the 2030 result, it is reassuring that the 2010's age distributions fit well to data. Thank you for the additional model validation efforts.

[LINK]

---

## [Editor Report · Decision Letter 4]

22 Nov 2023

Dear Dr Althoff, 

On behalf of my colleagues and the Academic Editor, Professor Marie-Louise Newell, I am pleased to inform you that we have agreed to publish your manuscript "The forecasted prevalence of comorbidities and multimorbidity in people with HIV in the United States through the year 2030: A modeling study" (PMEDICINE-D-22-03963R4) in PLOS Medicine.

Prior to publication please ensure you address the following:

1) Please cite and label your figures (and tables) in the main manuscript as outlined here https://journals.plos.org/plosmedicine/s/figures#loc-how-to-submit-figures-and-captions

2) Please cite and label your supporting information as outlined here

https://journals.plos.org/plosmedicine/s/supporting-information

PRESS

Best wishes,

Pippa 

Philippa Dodd, MBBS MRCP PhD 

PLOS Medicine